# Mathematical modeling of multiple pathways in colorectal carcinogenesis using dynamical systems with Kronecker structure

**Saskia Haupt**[1,2]*, **Alexander Zeilmann**[3], **Aysel Ahadova**[4], **Hendrik Bläker**[5], **Magnus von Knebel Doeberitz**[4], **Matthias Kloor**[4], **Vincent Heuveline**[1,2]

**1** Engineering Mathematics and Computing Lab (EMCL), Interdisciplinary Center for Scientific Computing (IWR), Heidelberg University, Heidelberg, Germany, **2** Data Mining and Uncertainty Quantification (DMQ), Heidelberg Institute for Theoretical Studies (HITS), Heidelberg, Germany, **3** Image and Pattern Analysis Group (IPA), Heidelberg University, Heidelberg, Germany, **4** Department of Applied Tumor Biology (ATB), Institute of Pathology, University Hospital Heidelberg, Heidelberg, Germany, **5** Institute of Pathology, University Hospital Leipzig, Leipzig, Germany

* saskia.haupt@uni-heidelberg.de

**Data Availability Statement:** All relevant data are within the manuscript and its Supporting information files.

## Abstract

Like many other types of cancer, colorectal cancer (CRC) develops through multiple pathways of carcinogenesis. This is also true for colorectal carcinogenesis in Lynch syndrome (LS), the most common inherited CRC syndrome. However, a comprehensive understanding of the distribution of these pathways of carcinogenesis, which allows for tailored clinical treatment and even prevention, is still lacking. We suggest a linear dynamical system modeling the evolution of different pathways of colorectal carcinogenesis based on the involved driver mutations. The model consists of different components accounting for independent and dependent mutational processes. We define the driver gene mutation graphs and combine them using the Cartesian graph product. This leads to matrix components built by the Kronecker sum and product of the adjacency matrices of the gene mutation graphs enabling a thorough mathematical analysis and medical interpretation. Using the Kronecker structure, we developed a mathematical model which we applied exemplarily to the three pathways of colorectal carcinogenesis in LS. Beside a pathogenic germline variant in one of the DNA mismatch repair (MMR) genes, driver mutations in *APC*, *CTNNB1*, *KRAS* and *TP53* are considered. We exemplarily incorporate mutational dependencies, such as increased point mutation rates after MMR deficiency, and based on recent experimental data, biallelic somatic *CTNNB1* mutations as common drivers of LS-associated CRCs. With the model and parameter choice, we obtained simulation results that are in concordance with clinical observations. These include the evolution of MMR-deficient crypts as early precursors in LS carcinogenesis and the influence of variants in MMR genes thereon. The proportions of MMR-deficient and MMR-proficient APC-inactivated crypts as first measure for the distribution among the pathways in LS-associated colorectal carcinogenesis are compatible with clinical observations. The approach provides a modular framework for modeling multiple pathways of carcinogenesis yielding promising results in concordance with clinical observations in LS CRCs.

**Funding:** The authors received no specific funding for this work.

**Competing interests:** The authors have declared that no competing interests exist.

## Author summary

Cancer is a disease caused by alterations of the genome. The alterations can affect each component of the genome, whereas only some lead to a change in the functioning of the cell. As there are several of those so-called driver mutations, there are different possibilities in which order they can occur. It is currently assumed that the order of driver mutations is linked to the course of cancer and thus to clinical treatment and even prevention. However, cells with a driver mutation, which carry a risk to grow out to a tumor, are clinically invisible for a long time. This means the early carcinogenesis is a hidden process. Mathematical models allow testing related medical hypotheses to obtain a better understanding of the underlying biological processes. We proposed a mathematical model for different molecular pathways of carcinogenesis based on a linear dynamical system. Thereby, we used the Kronecker structure, a specific structure which allows for a thorough mathematical analysis and medical interpretation. The model consists of multiple components to account for independent and dependent mutational processes. For the presented work, we focused on cancer development in the colon. However, modifications of the model could be applied to other organs.

## 1 Introduction

Cancer is the second leading cause of death worldwide accounting for an estimated 9.6 million deaths in 2018, whereby the second most common type is colorectal cancer (CRC) [1]. Still, adequate treatment and in particular prevention strategies are lacking in many cases, as it is difficult to investigate the process of cancer development, called carcinogenesis, right from the beginning.

In this work, we present a mathematical model of colorectal carcinogenesis. It takes into account the multiple pathway nature of carcinogenesis (Fig 1A) reflecting different types of CRC based on molecular parameters with individual needs for prevention and treatment [2].

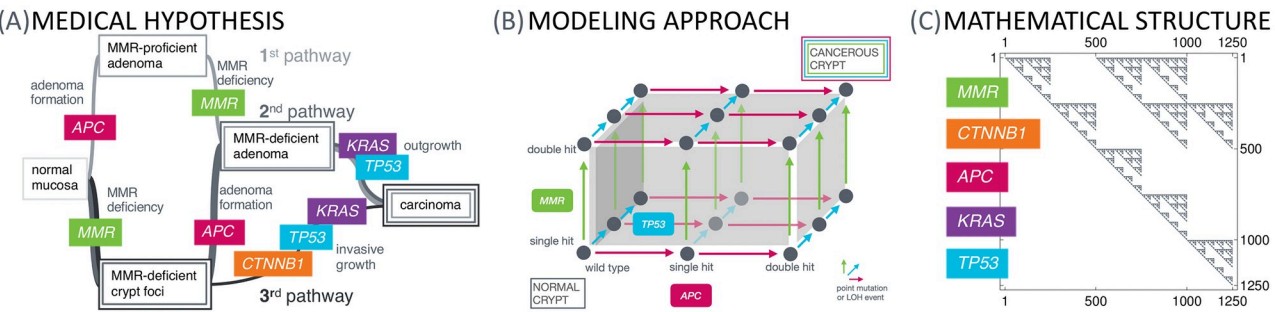

**Fig 1. From the medical hypothesis over the modeling approach to the mathematical structure.** The medical hypothesis of multiple pathways of carcinogenesis is widely known for various types of cancer. (A) We present a model for this phenomenon at the example of Lynch syndrome, the most common inherited CRC syndrome, with specific key driver events in the MMR genes, *CTNNB1*, *APC*, *KRAS* and *TP53*. (B) This current medical understanding of carcinogenesis is translated into a mathematical model using a specific dynamical system, which can be represented by a graph structure, where each vertex in the graph represents a genotypic state and the edges correspond to the transition probabilities between those states. Starting with all colonic crypts in the state of all genes being wild-type and a single MMR germline variant due to Lynch syndrome, we are interested in the distribution of the crypts among the graph at different ages of the patient in order to obtain estimates for the number of crypts in specific states, e.g., adenomatous or cancerous states. (C) The underlying matrix of the dynamical system makes use of the Kronecker sum and product. It is a sparse upper triangular matrix accounting for the assumption that mutations cannot be reverted. This allows fast numerical solving by using the matrix exponential. Each nonzero entry of the matrix represents a connection between genotypic states in the graph.

The mathematical model makes use of a dynamical system with a specific matrix structure using Kronecker products and sums (Fig 1C) in order to systematically describe the mutational events of individual genes (Fig 1B). These mutational events can be independent of or depending on other mutations, accounting for different types of mutations and for currently available data.

To exemplify this approach, we build the model for Lynch syndrome, the most common inherited CRC syndrome with an estimated population frequency of 1 in 180 [3]. Lynch syndrome is associated with an inherited mismatch repair (MMR) gene variant [4]. CRCs which develop in the context of Lynch syndrome mostly are MMR-deficient and enhance microsatellite instability (MSI) [5].

In addition to Lynch syndrome colorectal carcinogenesis, we modify the ansatz to model the sporadic counterpart of Lynch syndrome, often called Lynch-like cancers [6], as well as the classical adenoma-carcinoma sequence first described by Vogelstein and Kinzler [7] for microsatellite-stable (MSS) CRCs. Further, we apply the model to another hereditary CRC syndrome, familial adenomatous polyposis (FAP) [8].

## 1.1 Organization

To make this paper self-contained, we elucidate the medical background in Section 1.2. Section 2 presents related work and our contribution in this context. The mathematical model is presented in Section 3.1 which is based on different components: The first model component implements independent mutational processes and the other components model known mutational dependencies. Section 3.2 represents modifications for non-Lynch scenarios or cancer in other organs than the colon. Section 4 demonstrates a selection of the results which can be obtained with the model and its modifications. Finally, we conclude in Section 5 discussing the assumptions of the model and their implications. For a mathematical background, we refer to S1 Appendix.

## 1.2 Medical background

Cancer is a disease caused by alterations of the genome, the carrier of genetic information [9, 10]. Precisely defining these changes, which are required to transform a normal cell of the human body into a malignant cancer cell, is a crucial step towards understanding the development of cancer.

**Multiple pathways of carcinogenesis.** In the early stages of cancer research, it was unknown whether the development of cancer, a process called carcinogenesis, was a purely chaotic process of random mutations. However, in 1959, Nowell and Hungerford [11] made the observation of a specific recurrent alteration across different cancers of the same type. This suggested the existence of at least a certain degree of order in the chaos.

In the following decade, evidence emerged that one single mutation is normally insufficient to drive a cell into malignancy because cells possess multiple control mechanisms which protect the organism from the uncontrolled growth of single cells. Thus, Vogelstein, Fearon and Kinzler [7, 12] established a step-wise hypothesis of cancer formation in the colon postulating that several mutations are required for the development of cancer cells. This Adenoma-Carcinoma Hypothesis describes the formation of certain precancerous lesions and their progression into a manifest cancer. The model implies that adenomas are the precursor lesions of most colorectal cancers and it describes typical molecular events associated with progression to cancer. The step-wise hypothesis has been validated subsequently in many independent studies for many different cancer types. Currently, it is expected that a minimum number of three mutation events is required to transform a normal cell into a cancer cell. This hypothesis is called the three strikes hypothesis [13]. Accordingly, cancer for the present modeling

approach is defined as a state, in which alterations of at least three key signaling pathways or respective genes are present in one crypt (see also Section 4).

Mutations occur over the whole genome, whereby we differentiate between two broad classes: So-called point mutations only affect a single nucleotide, while loss of heterozygosity (LOH) refers to the loss of some region in one copy of the diploid genome, which can result in the deletion of whole genes.

If mutations strike in regions with a protein-encoding function, two main scenarios that can favor uncontrolled cell growth are seen: Somatic mutations can either directly activate oncogenes (typically referred to in the literature as gain-of-function mutations), which physiologically promote appropriate cell growth and proliferation, through conformational changes or impairing self-inactivation, or mutations can damage or destroy tumor suppressor genes (typically referred to in the literature as loss-of-function mutations), which physiologically limit cell growth and proliferation.

These coding mutations have to be identified from all the possible mutations that can occur, as they might have a functional impact on the cell. This includes the identification of oncogenes and tumor suppressor genes, but there are many more mutations to be identified. Moreover, only a certain combination of these mutations will lead to cancer in the end. This might be due to the fact that some mutations have a growth-repressing effect and lead to cell death. Further, there is the possibility of controlling cancer by non-cell autonomous mechanisms, like immune surveillance, which is especially important for the presented example of Lynch syndrome [14]. Apart from that, current data raise the possibility that the immune system may not only remove precursor lesions but also infiltrating cancers, as described for Lynch syndrome-associated cancers [15].

Different combinations of key mutations result in several distinct pathways to be distinguished by the involved genes and the ordering thereof. An important goal in cancer research is to investigate which of these pathways can arise in human carcinogenesis. Here, Lynch syndrome colorectal carcinogenesis is a prime example with three currently hypothesized main pathways of carcinogenesis [16] (Fig 1A) which will be explained in more detail in the next paragraph.

**Lynch syndrome-associated colorectal carcinogenesis.** Individuals with Lynch syndrome are predisposed to developing certain malignancies with a substantially higher lifetime risk compared to the general population. The most common Lynch syndrome manifestations are CRC (50% [17] compared to 6% in the normal population) and endometrial cancer (40–60% compared to 2.6% in women without Lynch syndrome) [4, 18]. Further, individuals have an increased lifetime risk for many other types of cancer such as in the stomach, small bowel, brain, skin, pancreas, biliary tract, ovary (only for women) and upper urinary tract [19].

Lynch syndrome carriers have an inherited pathogenic variant in one allele of the affected MMR genes *MLH1*, *MSH2*, *MSH6* or *PMS2* [20] passed down in the family from parent to child. Upon the second somatic hit inactivating the remaining allele, MMR deficiency manifests in the affected cell [21]. DNA replication errors, especially those which occur at repetitive sequences (microsatellites consisting of a consecutive series of identical basepairs) cannot be corrected by the mismatch repair system. MMR deficiency leads to microsatellite instability.

MMR deficiency can be an initiating or a secondary event in Lynch syndrome carcinogenesis. This is reflected by the hypothesis of three pathways responsible for colorectal carcinogenesis in Lynch syndrome [22] (see Fig 1): One pathway of carcinogenesis starts with adenoma formation, then MMR deficiency and cancer outgrowth; the second is initiated by MMR deficiency, then adenoma formation and cancer outgrowth; and the third shows MMR deficiency as initiating event and invasive cancer growth.

The relative proportion of one or the other pathway of carcinogenesis and the contribution of certain molecular events is thereby an open question with clinical implications: Ahadova et al. [16] showed that the molecular pathways of carcinogenesis are linked to different mutational processes, e.g., *CTNNB1*-mutant colorectal carcinomas are associated with immediate invasive growth, following the third presented pathway. Recent independent studies (analyzed in [23]) demonstrated that a substantial proportion of Lynch syndrome individuals develops CRC despite regular colonoscopy and that there is no difference in CRC incidence or stage at detection by colonoscopy with respect to different Lynch syndrome surveillance intervals [24]. This emphasizes the need for improved cancer prevention depending on the molecular footprints of carcinogenesis for Lynch syndrome individuals. Further, there are MMR gene-dependent differences regarding the risk of colorectal adenomas and carcinomas, and regarding somatic mutations in patients with Lynch syndrome [25] which supports the need of adjusting surveillance guidelines based on MMR gene variants.

As a special case of CRC, Lynch syndrome-associated colorectal cancer is widely believed to originate in colonic crypts [26]. Those are found in the epithelia of the colon and consist of different cell types [27], among others, stem cells located at the crypt base. They are important for tissue renewal due to their unlimited proliferative potential, however also prone to mutations. If a cell in a crypt becomes mutated, this mutation has to spread within the crypt such that the whole crypt is mutated and can be measured with current techniques, a process called fixation or monoclonal conversion [28]. Modeling this process and analyzing the role of colonic stem cells located at the crypt base is important to understand the intra-crypt dynamics. We are currently working on these aspects with first results in [29]. However, for the present model, we focus on the evolution of genetic states within crypts as a whole and compare the modeling results with currently available biological and epidemiological data.

## 2 Related work

First attempts to build mathematical models in cancer research were made in the middle of the 20th century. Armitage and Doll [30, 31] proposed and analyzed one of the first multistage models of carcinogenesis, which are based on the hypothesis that there are multiple subsequent steps before a cancer is formed. The model was extended in the following years [32, 33]. Among the first to consider a model of multiple pathways of carcinogenesis were Tan et al. [34, 35]. These are based on the hypothesis that there are several possible ways in which cancer can develop.

With the increasing medical knowledge about cancer development, it became more and more evident that a single model describing the whole process of carcinogenesis from the genomic, over the cell, up to the tissue, organ and organism-level is too complex to build. Nowadays, there exist different types of models describing individual aspects of carcinogenesis (in an unordered list of example publications):

▷ Modeling **healthy tissue formation**, such as the evolution of colonic crypts [36–38],

▷ detecting **driver genes** [39–42],

▷ estimating the most likely **temporal order of key mutations** [13, 43],

▷ modeling the **cancer-immune system interaction**, including neoantigen presentation [44–46],

▷ predicting **effects of intervention strategies** on tumor growth and patient survival, such as the effect of screening on adenoma risk [47].

From a mathematical point of view, the modeling makes use of different approaches, such as ordinary differential equations [48, 49], partial differential equations [50], stochastic processes [51, 52], graph theory [53–55], and statistics [56, 57].

For hereditary CRCs, in particular, Komarova et al. [48, 58] proposed a model for the occurrence and ordering of key events during carcinogenesis based on ordinary differential equations [48, 58], which was adapted to sporadic carcinogenesis. In particular, it addresses the question of the extent of genetic instability as an early event in carcinogenesis.

A recent paper by Paterson et al. [59] presents a model for quantifying the evolutionary dynamics of CRC initiation and progression based on describing the occurrence of key driver mutations. The individual mutational graphs of *APC*, *KRAS* and *TP53* in our model correspond to those in [59], considering *APC* and *TP53* as classical tumor suppressor genes and *KRAS* as classical oncogene in CRC. In addition, the general approach of calculating gene-specific numbers of driver positions as well as assuming *APC* and *KRAS* provide fitness advantage but not *TP53* are in concordance with [59]. The latter assumption is based on several independent studies [28, 37, 60].

## 2.1 Contribution

We provide a general mathematical framework that describes arbitrarily complex and arbitrary numbers of pathways and mutations because the chosen Kronecker structure enables a modular construction and an analytic, computationally efficient solution. We use Lynch syndrome carcinogenesis to illustrate the flexibility of the model. Naturally, specific assumptions may vary for other types of cancer. We illustrated model modifications for FAP, Lynch-like and the classical colorectal carcinogenesis.

Instead of focusing on modeling *APC* inactivation and MMR deficiency as in [48], we choose a more general approach for combining mutations in different genes. Compared to [59], we take into account different modes of cancer evolution beside the classical adenoma-carcinoma sequence of colorectal carcinogenesis, including hereditary forms like Lynch syndrome and familial adenomatous polyposis (FAP). Further, recent data show that in Lynch syndrome-associated CRCs, biallelic mutations of *CTNNB1* seem to be required to mediate an oncogenic driver effect [61, 62], which we included in the definition of the gene mutation graphs.

While the approach in [59] is a hybrid approach of linear ordinary differential equations (ODEs) and a stochastic branching process, we use a system of ODEs to model the evolution of all genotypic states which eases the computational solution process tremendously. This goes in hand with the fact that all formulas in our model are exact from a mathematical point of view without using any approximations which in turn allows for an analytical solution of the ODEs by using the matrix exponential.

Further, the model consists of different components for modeling independent and dependent mutational processes taking into account currently available clinical observations and biomedical data.

Finally, our approach makes it possible to easily include new medical insights, while preserving the other properties of the model, like the integration of the involved differential equations. This incorporates the possibility for multiple cancerous genotypic states reflecting the real world heterogeneity of cancer, the consideration of multiple driver genes, as well as the use of different initial values and parameter combinations for modeling other carcinogenesis processes.

## 3 Methods

### 3.1 Modeling Lynch syndrome carcinogenesis

In this section, we introduce our model for colorectal carcinogenesis in Lynch syndrome. The model consists of a dynamical system given in the form of a linear ordinary differential equation which is constructed with the help of adjacency matrices describing the joint process of mutations in several genes, including mutations independent of and depending on other mutations. All mutations are assumed to be present in the whole crypt. Mutations which occur in one cell but are washed out as they reach the top of the crypt and undergo apoptosis are not considered in the model.

The system matrix is built in an additive way for implementing independent and dependent mutational processes. The matrix $A$ for the independent processes is based on three main assumptions leading to the Kronecker sum in a natural way: 1) All combinations of mutations in the considered genes are possible and there are no additional genotypic states, 2) no two mutations in different genes occur at the exactly same point in time, 3) the mutational processes are independent of each other (see also Section 2 in S1 Appendix).

The model components representing dependent mutations are constructed in a similar way using the Kronecker structure, but here we do not make the assumptions 2 and 3. This allows for modeling dependent mutations and for the possibility of simultaneous mutations (see model components $B$, $C$, $D$, $E$ and $F$).

**3.1.1 Gene mutation graphs.**   In the case of colorectal carcinogenesis in Lynch syndrome, the MMR gene mutations are are associated with an increased cancer lifetime risk of Lynch syndrome individuals. Besides the MMR genes, we consider four additional possible driver genes, namely *APC*, *KRAS*, *CTNNB1* and *TP53* which are typical representatives of the oncogenes and tumor suppressor genes affected in the corresponding pathways of Lynch syndrome-associated carcinogenesis.

Each of these genes can have a variety of mutation status:

**State $\emptyset$:** In this state, none of the alleles has a point mutation or is affected by an LOH event.

**States m and mm:** These states describe one allele being hit by a point mutation (where the other one is not mutated) and point mutations on both alleles.

**States l and ll:** Similarly, these states describe one (respectively two) allele(s) being affected by an LOH event.

**State ml:** One of the alleles has obtained a point mutation and in the other one, an LOH event occurred. We do not differentiate which allele has which mutation and in which order they happened.

We assume that ll in *CTNNB1*, *APC* and *TP53* damage a cell in such a way that it directly leads to cell death [59]. Thus, there will be no crypt with all cells being in that state. As we model the evolution of genotypic states of crypts, we do not consider the ll status for *CTNNB1*, *APC* and *TP53*.

As our example is Lynch syndrome carcinogenesis, all cells and hence, also all crypts have a single germline variant in the respective MMR gene and there is no $\emptyset$ status for MMR.

Further, *APC* and *TP53* are tumor suppressor genes meaning that both alleles have to be mutated for an inactivation, whereby this two hit hypothesis dates back to Knudson et al in 1971 [63]. In particular, we ignore a possibly dominant-negative effect of *APC* and *TP53* mutations resulting in a single hit necessary for inactivation [64].

In addition, *KRAS* is an oncogene, where one activating mutation is necessary. In Lynch syndrome-asscociated CRC, biallelic mutations of *CTNNB1* seem to be required to mediate an oncogenic driver effect [61, 62].

All these assumptions lead to the vertex sets

$$\mathcal{V}_{\text{MMR}} = \{\text{m}, \text{l}, \text{mm}, \text{ml}, \text{ll}\}, \tag{1}$$

$$\mathcal{V}_{\text{CTNNB1}} = \{\emptyset, \text{m}, \text{l}, \text{mm}, \text{ml}\}, \tag{2}$$

$$\mathcal{V}_{\text{APC}} = \{\emptyset, \text{m}, \text{l}, \text{mm}, \text{ml}\}, \tag{3}$$

$$\mathcal{V}_{\text{KRAS}} = \{\emptyset, \text{m}\}, \tag{4}$$

$$\mathcal{V}_{\text{TP53}} = \{\emptyset, \text{m}, \text{l}, \text{mm}, \text{ml}\}. \tag{5}$$

Using these vertex sets, we construct gene mutation graphs, in which we connect the mutation status that differ by only one mutation. This means we assume that only one mutation happens at any specific time point.

Further, we make the assumption that once a mutation has happened it cannot be reversed by another mutation. Because of this, the mutation graphs are directed acyclic graphs and their adjacency matrix can be written as a triangular matrix.

The resulting graphs are illustrated in Fig 2. This figure also displays the edge weights of the gene mutation graph, i.e., the likelihood that we transfer from one mutation status to another. The choice of the edge weights will be explained in the following sections.

**3.1.2 Point mutations.** To model the likelihood $p_{\text{pt}}(\text{gene})$ for crypts being affected by point mutations in a specific gene, we make the following configurable assumptions for the example of Lynch syndrome colorectal carcinogenesis. For other types of cancer, or once new medical insights are gathered, they can and should be adapted.

▷ We would like to model the evolution of crypts over years. Many measurements and estimates are given in days. Thus, we use the factor 365 to convert the measurements per day to measurements per year.

▷ In each cell division, we accumulate $n_{\text{pt}} = 1.2$ point mutations according to measurements in [65], where we assume that a cell division takes one day [27].

▷ The point mutations are uniformly distributed over the base pairs on the entire genome.

▷ Each crypt is estimated [37] to consist of approximately $1.7 \cdot 10^3$ to $2.5 \cdot 10^3$ cells, whereas only approximately 75% of them can divide. Thus, we use $n_{\text{cells}} = 1500$ as an approximation to the number of cells per crypt.

▷ There are $n_{\text{bp,genome}} = 3.2 \cdot 10^9$ base pairs (bp) on the genome.

▷ Only the point mutations which occur in hotspots of the genes are relevant for cancer development. Hotspots are regions of a gene which give rise to a phenotypical change if mutated. The size of the hotspots $n_{\text{hs}}(\text{gene})$ is gene dependent and is explained in the following.

▷ Not all point mutations which appear in a crypt take over the entire crypt [28]. We model this in a gene dependent fixation affinity $f(\text{gene})$, i.e., the tendency of a cell with a mutation in a gene to take over the whole crypt.

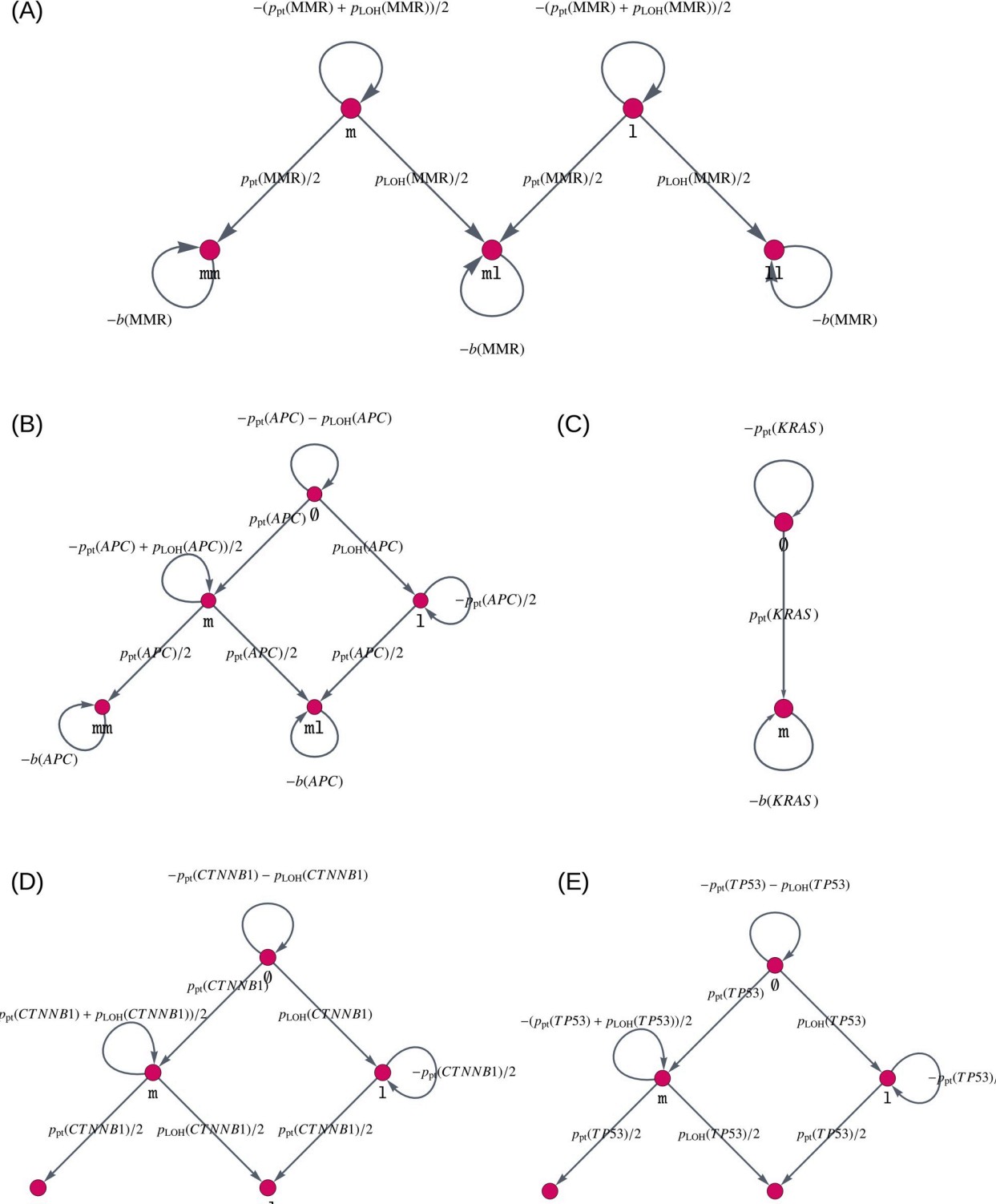

**Fig 2. Gene mutation graphs for independent mutational processes.** These graphs represent the possible mutation status, i.e., which mutations the alleles of the gene can have accumulated, as vertices $\emptyset$, m, l, mm, ll and ml. They are given for (A) MMR gene mutations, (B) *APC* mutations, (C) *KRAS* mutations, (D) *CTNNB1* mutations, and (E) *TP53* mutations. The edges connecting different vertices represent mutations, whereas self-loops, i.e., edges that connect a vertex with itself, describe no mutation occurring at the current point in time. The edges are labeled by the amount of change which happens at each point in time. Note that in the colon, biallelic mutations of *CTNNB1* seem to be required to mediate an oncogenic driver effect [61, 62], leading to a gene mutation graph similar to that of *APC* and *TP53*.

▷ We assume that the alleles are independent of each other, i.e., a mutation in one allele does not influence the mutation probability in the other allele. Thus, the likelihood $p_{\text{pt}}(\text{gene})$ is twice as large if there is no mutated allele ($n_{\text{mut}}(\text{gene}) = 0$) compared to the state where one allele is already mutated ($n_{\text{mut}}(\text{gene}) = 1$).

These assumptions lead to the following formula for the likelihood $p_{\text{pt}}(\text{gene})$:

$$p_{\text{pt}}(\text{gene}) = 365 \; n_{\text{pt}} \; n_{\text{cells}} \; \frac{n_{\text{hs}}(\text{gene})}{n_{\text{bp,genome}}} \; f(\text{gene}) \left( 1 - \frac{1}{2} n_{\text{mut}}(\text{gene}) \right). \tag{6}$$

Regarding the hotspots, we assume for *MLH1*, *MSH2* and *TP53* that the whole coding sequence is susceptible to inactivating point mutations, where we use the reference sequence database at NCBI for coding sequence lengths [66]. For *APC*, we use mutation data from the publicly available DFCI database using the cBioPortal website [67, 68]. We make use of data from about 4000 CRC samples to identify approximately 2400 hotspots.

For the present parameter choice, we assume for *CTNNB1* that only 5 mutations in codon 45 are relevant, according to [16]. Further, for *KRAS*, we assume 7 relevant mutations [22]. In summary, we obtain the following numbers for $n_{\text{hs}}$ given in Table 1.

**3.1.3 LOH events.**   We assume that all detectable LOH events are large enough to inactivate an affected gene. In other words, we assume that if LOH affects a certain gene, then an exon will be lost and the gene, therefore, is inactivated. As a consequence, the probability of LOH $p_{\text{LOH}}(\text{gene})$ for a given gene is proportional to its length, denoted by $n_{\text{bp}}(\text{gene})$.

The probability of a relevant LOH event for a specific gene with $n_{\text{mut}}(\text{gene}) \in \{0, 1, 2\}$ already mutated alleles and length $n_{\text{bp}}(\text{gene})$bp to be present in the whole crypt is given by

$$p_{\text{LOH}}(\text{gene}) = 365 \; n_{\text{cells}} \left( 1 - \frac{1}{2} n_{\text{mut}}(\text{gene}) \right) \; \alpha \; n_{\text{bp}}(\text{gene}) \; f(\text{gene}), \tag{7}$$

where $\alpha \in \mathbb{R}_{>0}$ is a parameter to be estimated, independent of the considered gene.

The available data for *MLH1* suggests that inactivation is twice as likely to occur due to LOH than due to point mutations [69]. Thus, we assume

$$p_{\text{LOH}}(\textit{MLH1}) \quad = 2 \; p_{\text{pt}}(\textit{MLH1}). \tag{8}$$

**Table 1. Estimates for $n_{\text{hs}}$.**

| gene | $n_{\text{hs}}$ |
|------|------|
| *MLH1* | 2,270 |
| *MSH2* | 2,800 |
| *CTNNB1* | 5 |
| *APC* | 2,400 |
| *KRAS* | 7 |
| *TP53* | 1,180 |

The given estimates are used for the computation of the point mutation rates for the individual genes. Those are based on the following data from the literature: *MLH1*, *MSH2* and *TP53*: [66]; *CTNNB1*: [16]; *APC*: [67, 68]; *KRAS*: [22].

Together with (6) and (7), we get

$$\alpha \quad = 2\frac{n_{\mathrm{hs}}(MLH1)}{n_{\mathrm{bp}}(MLH1)}\frac{n_{\mathrm{pt}}}{n_{\mathrm{bp,genome}}}. \tag{9}$$

In order to determine $\alpha$ and $p_{\mathrm{LOH}}$, we again use the reference sequence database at NCBI for the length of individual genes [66] given in Table 2.

**3.1.4 Fitness advantages and clonal expansion.** There is the possibility of introducing fitness changes $b$(gene) for individual mutation status of a gene. As we model the evolution of mutations at the crypt level, this corresponds to the clonal expansion of the crypts with one of the considered mutations. A fitness advantage is ensured by $b$(gene)$>$0 and a disadvantage with $b$(gene)$<$0. By using the notion of graphs, this corresponds to a self-loop of the respective genotypic state node with a weight equal to the fitness change. We assume that MMR deficiency leads to a fitness disadvantage [70], i.e., $b$(MMR)$<$0, and *APC* inactivation and *KRAS* activation lead to a fitness advantage, i.e., $b$(*APC*) $>$ 0 and $b$(*KRAS*) $>$ 0, in concordance with current measurements [28, 71].

In other words, the proliferation and disappearance of certain genotypic states is jointly modeled by the self-loops in the graph. This largely reduces the number of probability parameters necessary to be determined, accounting for the fact that there are currently not enough prospective data available to estimate or learn all the parameters. However, once there are enough data available, an additional state for dead or disappearing lesions can be introduced. We describe the corresponding formulas in S1 Appendix.

**3.1.5 A model for carcinogenesis.** Our mathematical model of multiple pathways in Lynch syndrome carcinogenesis is given by a system of linear ordinary differential equations

$$\dot{x}(t) = (A+B+C+D+E+F)^{\top}x(t), \quad x(0) = x_0. \tag{10}$$

The system matrix with its additive components implements the independent mutational processes in the matrix $A$ and all mutational dependencies, supported by available data, in the matrices $B$, $C$, $D$, $E$ and $F$. How the individual matrices are built mathematically is introduced in the following paragraphs.

We shortly explain how the model (10) is solved. While the system matrix has $1250 = 5\cdot5\cdot2\cdot5\cdot5$ rows and columns, corresponding to all possible genotypes, it is very sparse, as illustrated in Fig 3A.

The transpose of the matrix is merely due to different notation conventions for adjacency matrices and differential equations.

We assume that the Lynch syndrome individuals have no mutations at birth except for an MMR germline variant due to a point mutation (90–95% of individuals) or due to an LOH

**Table 2. Estimates for $n_{\mathrm{bp}}$.**

| gene | $n_{\mathrm{bp}}$ |
| --- | --- |
| *MLH1* | 57,500 |
| *MSH2* | 80,000 |
| *CTNNB1* | 41,000 |
| *APC* | 139,000 |
| *TP53* | 19,200 |

The following estimates for $n_{\mathrm{bp}}$ are necessary for the computation of the LOH rates for the individual genes. They are based on the reference sequence database at NCBI [66].

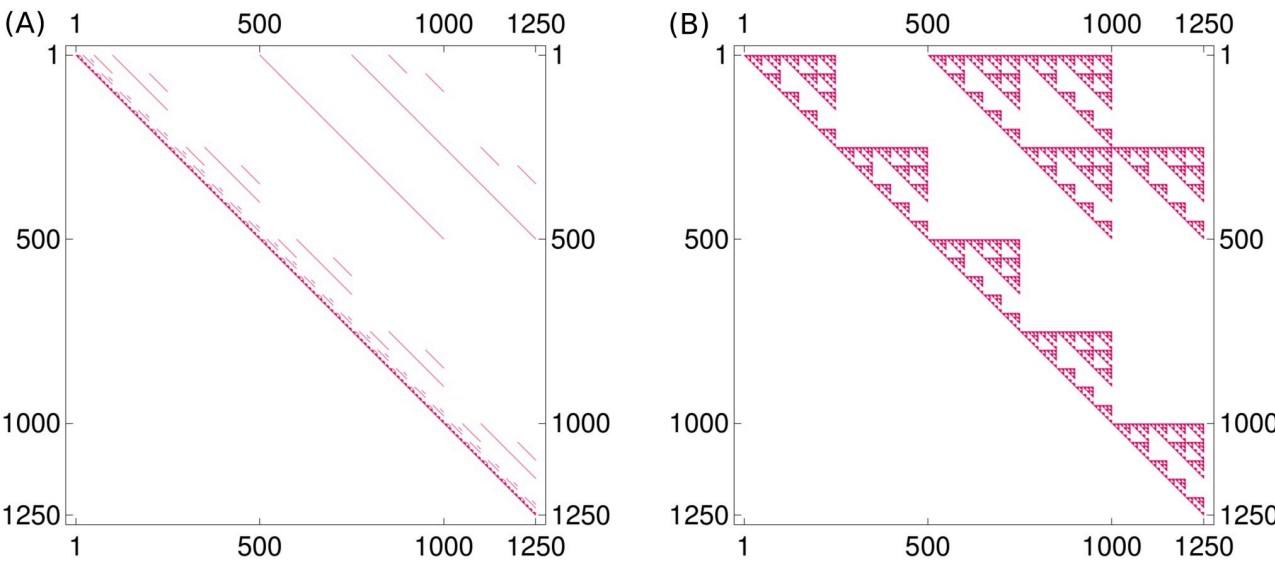

**Fig 3. Sparse matrix structure.** (A) The system matrix $(A + B + C + D + E + F)$ of the linear model is a very sparse matrix, i.e., only a few entries are nonzero. These nonzero entries are colored red in the plot, which also illustrates the fact that $(A + B + C + D + E + F)$ is an upper triangular matrix. (B) The sparsity structure of the matrix $\text{expm}(A + B + C + D + E + F)$, which is reminiscent of a Sierpiński fractal, is due to the individual matrices being the Kronecker product and sum of matrices. The two plots also illustrate nicely how modeling sparse local interactions in the matrix $(A + B + C + D + E + F)$ can have a more global effect in $\text{expm}(A + B + C + D + E + F)$.

event (5–10% of individuals) [72]. We differentiate these two groups of individuals by using different initial values for the differential equation. The initial value $x_0$ for the first group of individuals is

$$x_0 = n_{\text{crypts}} \; e_{\text{m}} \otimes \underbrace{e_\emptyset \otimes e_\emptyset \otimes e_\emptyset \otimes e_\emptyset}_{\substack{\text{no mutations in } CTNNB1, \\ APC, KRAS \text{ and } TP53}}, \tag{11}$$

where $n_{\text{crypts}} = 9.95 \cdot 10^6$ is the estimated [73] number of crypts in the colon and $e_{\text{m}}$ denotes the unit vector, which is zero everywhere, except for a 1 at the entry corresponding to the state m. This initial value can also be described as a vector which has the entry $n_{\text{crypts}}$ at the position corresponding to the genotype $(\text{m}, \emptyset, \emptyset, \emptyset, \emptyset)$ and is zero everywhere else.

Accordingly, the initial value for the second group of individuals is given by

$$x_0 = n_{\text{crypts}} \; e_1 \otimes \underbrace{e_\emptyset \otimes e_\emptyset \otimes e_\emptyset \otimes e_\emptyset}_{\substack{\text{no mutations in } CTNNB1, \\ APC, KRAS \text{ and } TP53}}. \tag{12}$$

As stated in Eq (S1–8) in S1 Appendix, the exact solution of the differential equation is given by $x(t) = \text{expm}(t(A + B + C + D + E + F)^\top)x_0$. We illustrate the sparsity structure of the matrix exponential in Fig 3B.

**Model component for independent mutations.** We explain how the matrix $A$ for independent mutational processes is built. Having defined the gene mutation graphs with adjacency matrices $A_{\text{MMR}}, A_{CTNNB1}, A_{APC}, A_{KRAS}, A_{TP53}$ for different genes (Fig 2), we combine them using the Kronecker product as explained in Section 2 in S1 Appendix. Accordingly, the adjacency matrix of the combined model is given by the Kronecker sum of the adjacency matrices

of the individual genes

$$A = A_{\text{MMR}} \oplus A_{CTNNB1} \oplus A_{APC} \oplus A_{KRAS} \oplus A_{TP53}. \tag{13}$$

When only considering independent mutational processes, the model (10) reduces to

$$x(t) = A^{\top} x(t), \quad x(0) = x_0, \tag{14}$$

where the solution can be rewritten in the following way (see Eq (S1–1) in S1 Appendix)

$$
\begin{aligned}
x(t) \quad &= \text{expm}(tA_{\text{MMR}}^{\top})e_{\text{m}} \otimes \text{expm}(tA_{CTNNB1}^{\top})e_{\emptyset} \otimes \text{expm}(tA_{APC}^{\top})e_{\emptyset} \\
&\otimes \ \text{expm}(tA_{KRAS}^{\top})e_{\emptyset} \otimes \text{expm}(tA_{TP53}^{\top})e_{\emptyset} n_{\text{crypts}}
\end{aligned}
\tag{15}
$$

for the case of the first group of individuals (11). This reduces the computational costs tremendously, as only several small matrices have to be considered instead of one large matrix.

**The model components for mutational dependencies.** The first model component, given by matrix $A$, implements all mutational processes that are independent of each other, which is either due to a independence indicated by data or due to missing medical insight suggesting otherwise. However, mutations change the functional behavior of a cell and thus, there are specific mutations that affect the probability of certain other mutations. In other words, there are mutations which are mutually exclusive or mutations which increase the probability of mutations in other genes [74].

Instead of changing the adjacency matrix $A$, we add the adjacency matrices for the dependent mutational processes to the independent one. This allows us to study the effects of the different mutational processes individually and makes it possible to include further dependencies when additional data are available in the future.

For the approach presented here, we assume and model the following molecular and biological mechanisms:

**Matrix $B$:** increased point mutation rate of *APC* after MMR deficiency,

**Matrix $C$:** positive association of *CTNNB1* and *MLH1* alterations,

**Matrix $D$:** increased LOH rate after *APC* inactivation,

**Matrix $E$:** mutual enhancement of effects $C$ and $D$,

**Matrix $F$:** increased mutation rate of *KRAS* after MMR deficiency.

In the following paragraphs, we explain all considered mutational dependencies in detail.

**Increased point mutation rate of *APC* after MMR deficiency.** MMR deficiency leads to an increased mutation rate, especially in microsatellites [20]. Among others, this is true for the point mutation rate of *APC*. Thus, we assume that the point mutation rate of *APC* is increased by a factor $\beta + 1$ if the crypt has an MMR-deficient state. This is assumed to be independent of the state of the other genes.

As we do not want to change the matrix $A$, we introduce an additional matrix $B$. This means, instead of multiplying single entries of $A$ by $\beta + 1$, we add a matrix $B$ to $A$ with corresponding entries multiplied by $\beta$.

We define the matrix $B$ by

$$B = B_{\text{MMR}} \otimes B_{CTNNB1} \otimes B_{APC} \otimes B_{KRAS} \otimes B_{TP53}, \tag{16}$$

where $B_{APC}$ is the adjacency matrix of the gene mutation graph in Fig 4 and

$$B_{\text{MMR}} = \text{diag}\,(0, 0, 1, 1, 1), \quad B_{CTNNB1} = I_5 = B_{TP53}, \quad B_{KRAS} = I_2. \tag{17}$$

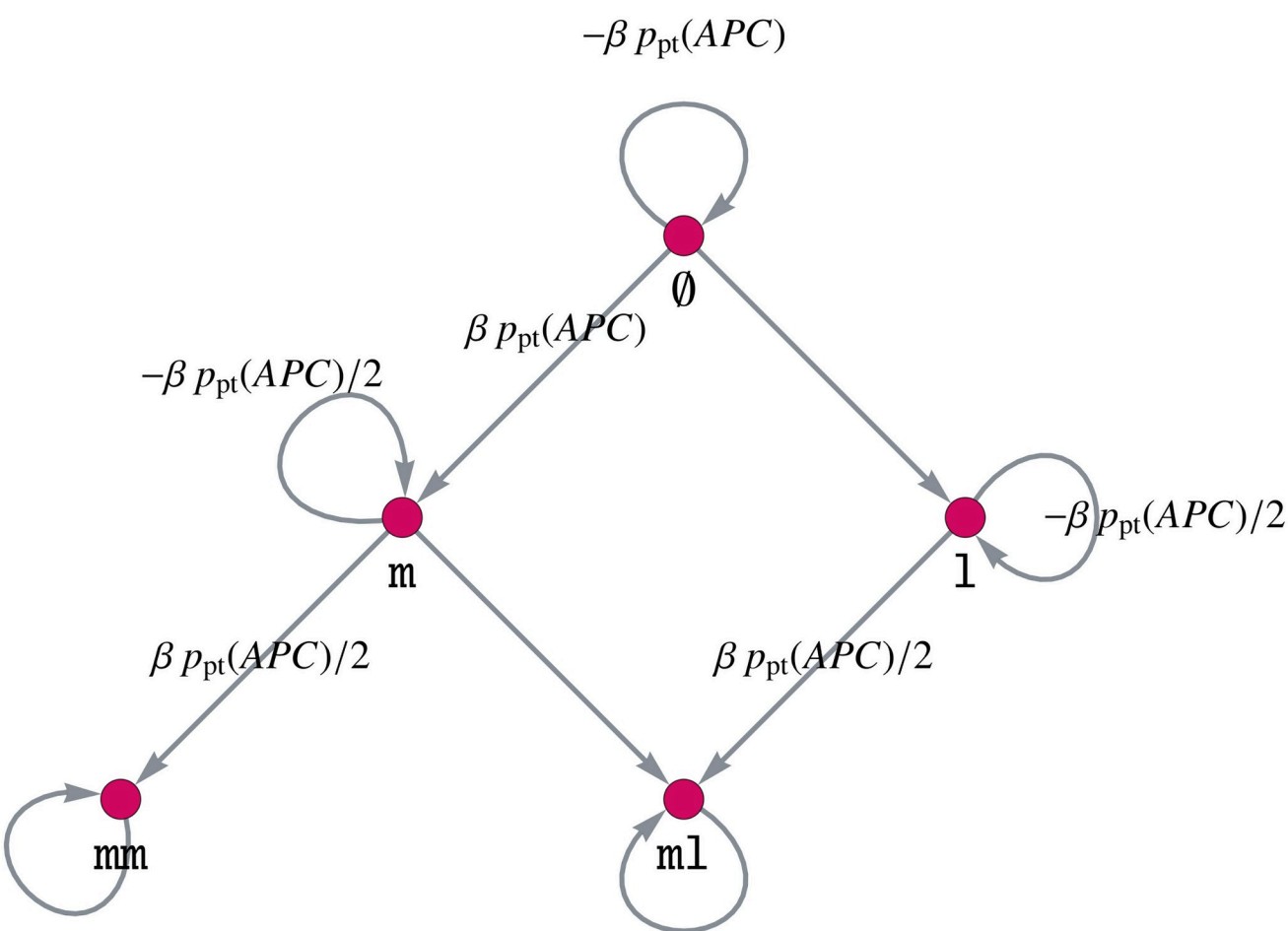

**Fig 4. Gene mutation graph of *APC* for increasing the point mutation rate of *APC* after MMR deficiency.**

Here, $\mathrm{diag}\,(d_1, d_2, \ldots, d_n) \in \mathbb{R}^{n \times n}$ denotes a diagonal matrix with entries $d_i$, $i \in \{1, 2, \ldots, n\}$ on its diagonal.

The definition (16) of the matrix $B$ yields the desired result of increasing the point mutation rate of *APC* after MMR deficiency. This can be explained intuitively: We only want to increase the point mutation rate after MMR deficiency, meaning that the MMR state is `mm`, `ml` or `ll`, leading to the matrix $B_{\mathrm{MMR}}$. Further, this influence of MMR on *APC* is independent of the other genes, meaning that it should hold for all states of the other genes. Thus, we choose the respective identity matrices for *KRAS*, *CTNNB1* and *TP53* and connect all matrices via the Kronecker product, instead of the Kronecker sum as in the matrix $A$.

**Positive association of *CTNNB1* and *MLH1* alterations.** According to [25], somatic *CTNNB1* mutations are significantly higher in *MLH1*-cancers than in the other MMR gene-associated CRCs. For illustration purposes, we make the assumption that inactivation of *MLH1* and *CTNNB1* are triggered by non-independent events. We calculate this dependency with an occurrence rate $r_{\mathrm{effLOH}}$, which we set to $r_{\mathrm{effLOH}} = 0.9$, and introduce an additional matrix $C$. The latter is based on a combined gene mutation graph for *MLH1* and *CTNNB1* and its connection with the remaining genes via the Kronecker product. Note that this is possible due to the chosen ordering of the genes.

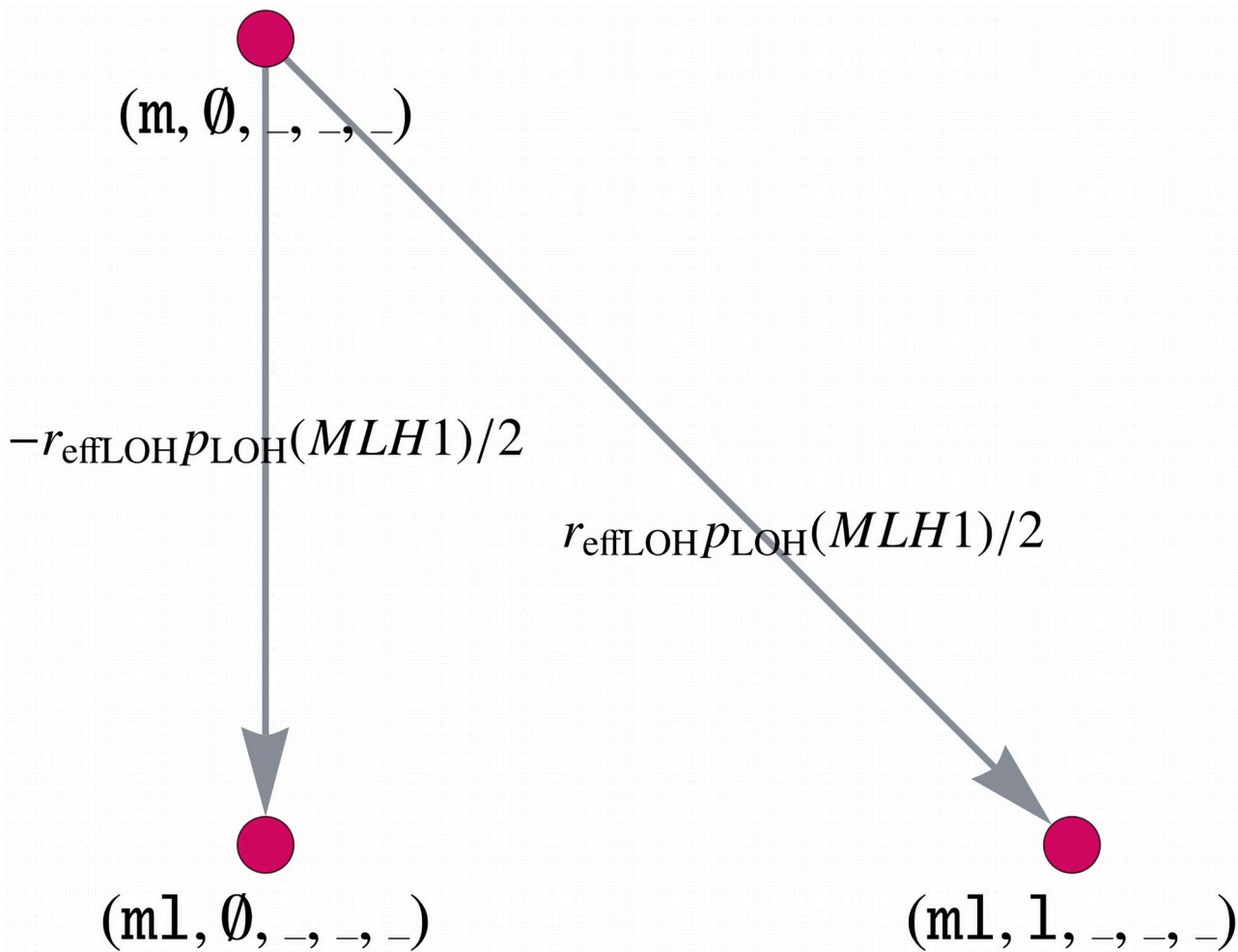

**Fig 5. Model component for the positive association of *MLH1* and *CTNNB1*.** Part of the combined gene mutation graph for *CTNNB1* and *MLH1* of the matrix *C*. The gene mutation graphs for the other possible gene states *MLH1* ∈ {l, ll}, *CTNNB1* ∈ {m, ml} are defined in an analogous way.

The matrix $C \in \mathbb{R}^{1250 \times 1250}$ is given by

$$C = C_{MLH1,CTNNB1} \otimes C_{APC} \otimes C_{KRAS} \otimes C_{TP53}, \tag{18}$$

where $C_{APC} = C_{TP53} = I_5$ and $C_{KRAS} = I_2$. The matrix $C_{MLH1,CTNNB1}$ is the adjacency matrix corresponding to the combined gene mutation graph for *MLH1* and *CTNNB1*. We explain in the following how this combined gene mutation graph is built and illustrate it in Fig 5.

Let _ denote an arbitrary state of the corresponding gene. Instead of multiplying the edge weight $p_{\mathrm{LOH}}(\mathrm{MMR})/2$ of the edge $(\mathrm{m}, \emptyset, \_, \_, \_) \to (\mathrm{ml}, \emptyset, \_, \_, \_)$ by $(1 - r_{\mathrm{effLOH}})$ in the original matrix *A*, we add a matrix *C* with a corresponding edge weight $-r_{\mathrm{effLOH}}\, p_{\mathrm{LOH}}(\mathrm{MMR})/2$. The

following edges are added to the matrix $C$ with the same weight:

$$(\mathtt{l}, \emptyset, \_, \_, \_) \quad \rightarrow (\mathtt{ll}, \emptyset, \_, \_, \_), \tag{19}$$

$$(\mathtt{m}, \mathtt{m}, \_, \_, \_) \quad \rightarrow (\mathtt{ml}, \mathtt{m}, \_, \_, \_), \tag{20}$$

$$(\mathtt{l}, \mathtt{m}, \_, \_, \_) \quad \rightarrow (\mathtt{ll}, \mathtt{m}, \_, \_, \_). \tag{21}$$

Furthermore, we need to insert the following new edges with edge weight $-r_{\text{effLOH}}$ $p_{\text{LOH}}(MLH1)/2$

$$(\mathtt{m}, \emptyset, \_, \_, \_) \quad \rightarrow (\mathtt{ml}, \mathtt{l}, \_, \_, \_), \tag{22}$$

$$(\mathtt{l}, \emptyset, \_, \_, \_) \quad \rightarrow (\mathtt{ll}, \mathtt{l}, \_, \_, \_), \tag{23}$$

$$(\mathtt{m}, \mathtt{m}, \_, \_, \_) \quad \rightarrow (\mathtt{ml}, \mathtt{ml}, \_, \_, \_), \tag{24}$$

$$(\mathtt{l}, \mathtt{m}, \_, \_, \_) \quad \rightarrow (\mathtt{ll}, \mathtt{ml}, \_, \_, \_). \tag{25}$$

All other entries of $C$ are zero, leading to a sparse matrix with only 400 non-zero entries.

**Increased LOH rate after *APC* inactivation.** The following model component deals with the increased LOH rate of *APC*-inactivated crypts, which is assumed to be the case in many cancers [52]. In the latter, we will denote those *APC*-inactivated crypts by *APC*-/-, which are inactivated due to $\mathtt{mm}$ or $\mathtt{ml}$.

As further LOH events can occur for MMR, *CTNNB1* and *TP53* in *APC*-/- crypts, we have to introduce individual matrices for each effect leading to the matrix $D = D_1 + D_2 + D_3$, where

$$D_1 = D_{\text{MMR}} \otimes I_5 \otimes \text{diag}\,(0, 0, 0, 1, 1) \otimes I_2 \otimes I_5, \tag{26}$$

$$D_2 = I_5 \otimes D_{\text{CTNNB1}} \otimes \text{diag}\,(0, 0, 0, 1, 1) \otimes I_2 \otimes I_5, \tag{27}$$

$$D_3 = I_5 \otimes I_5 \otimes \text{diag}\,(0, 0, 0, 1, 1) \otimes I_2 \otimes D_{TP53}. \tag{28}$$

Analogous to the model component $B$, we define a gene mutation graph of MMR, *CTNNB1* and *TP53* with parameter $\delta$ such that the LOH rate is increased by a factor $\delta + 1$. This is illustrated in Fig 6 for *CTNNB1* and *TP53*, where the gene mutation graph for MMR is defined analogously.

**Mutual enhancement of effects *C* and *D*.** *APC* inactivation increases the LOH rate of other genes, including *MLH1*, which is modeled by the matrix $D$. Further, there is a positive association of *MLH1* and *CTNNB1* alterations, which we can model in the same way as an LOH event, as described in matrix $C$. Thus, we would like to demonstrate how to model the mutual enhancement of two effects, which will be described by an additional matrix $E$. As for the matrix $C$, we build the combined adjacency matrix for *MLH1* and *CTNNB1* and combine it with the other genes via the Kronecker product, i.e.,

$$E = E_{MLH1,CTNNB1} \otimes \text{diag}\,(0, 0, 0, 1, 1) \otimes I_2 \otimes I_5, \tag{29}$$

where again, the ordering is essential to enable an efficient implementation.

This enhancement only affects the *APC*-/- crypts, thus we use diag(0, 0, 0, 1, 1) for the *APC* matrix. Analogous to Fig 5, we illustrate parts of the gene mutation graph for the combination of *MLH1* and *CTNNB1* after *APC* inactivation in Fig 7.

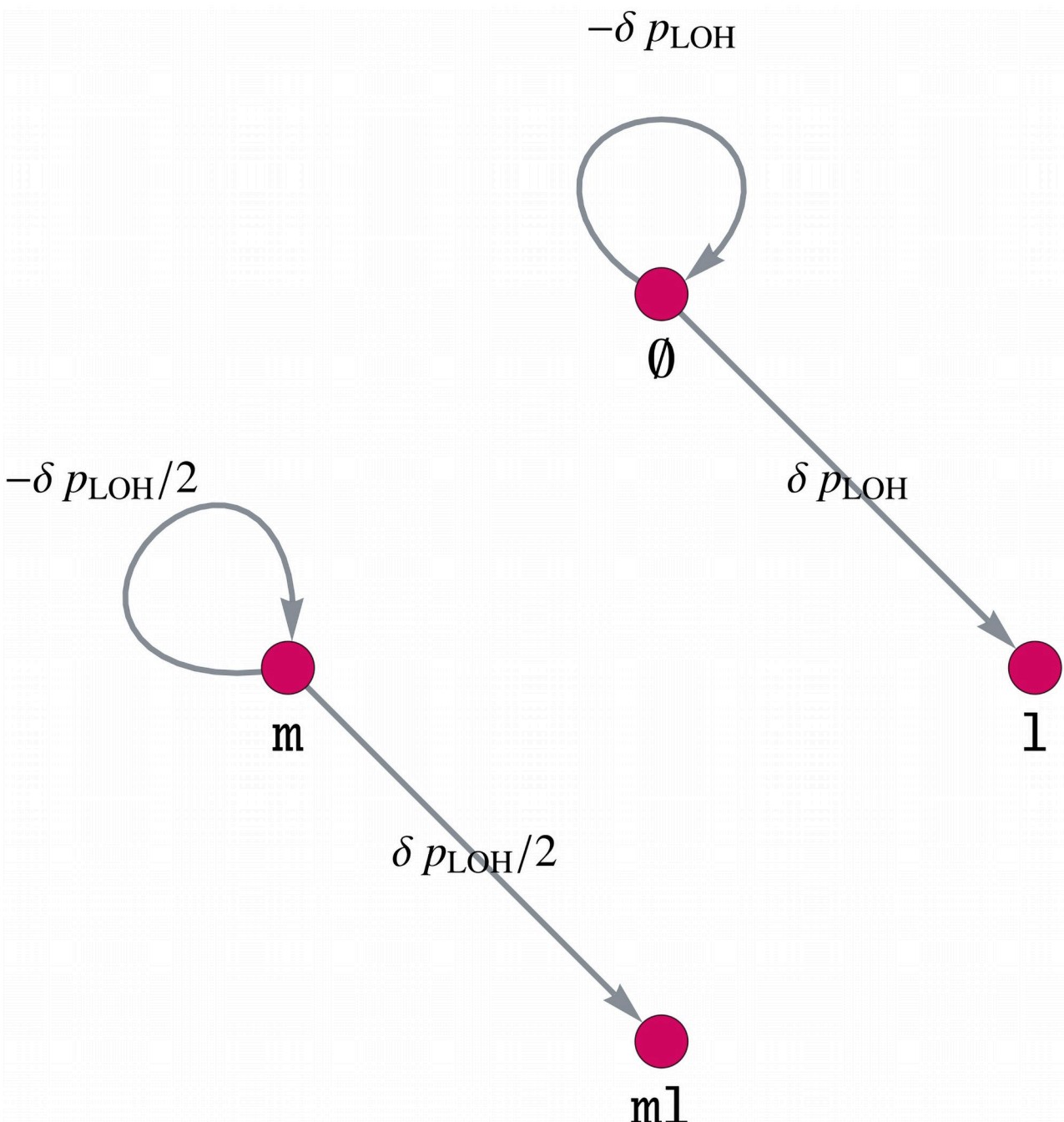

**Fig 6. Model component for increasing the LOH rate of MMR, *CTNNB1* and *TP53* by a factor $\delta$ + 1 after *APC* inactivation.** Gene mutation graph for both genes, *CTNNB1* and *TP53*, of the component *D*. The gene mutation graph for MMR is defined in an analogous way.

**Increased mutation rate of *KRAS* after MMR deficiency.** *KRAS* is an oncogene with one point mutation sufficient for activation, where mainly codon 12 or 13 are hit. Codon 13 mutations are known to be associated with and enriched in MMR-deficient cancers, as these mutations are more likely to occur under the influence of MMR deficiency [22]. We will consider this association by increasing the *KRAS* mutation rate after MMR deficiency by a factor $\zeta$ + 1.

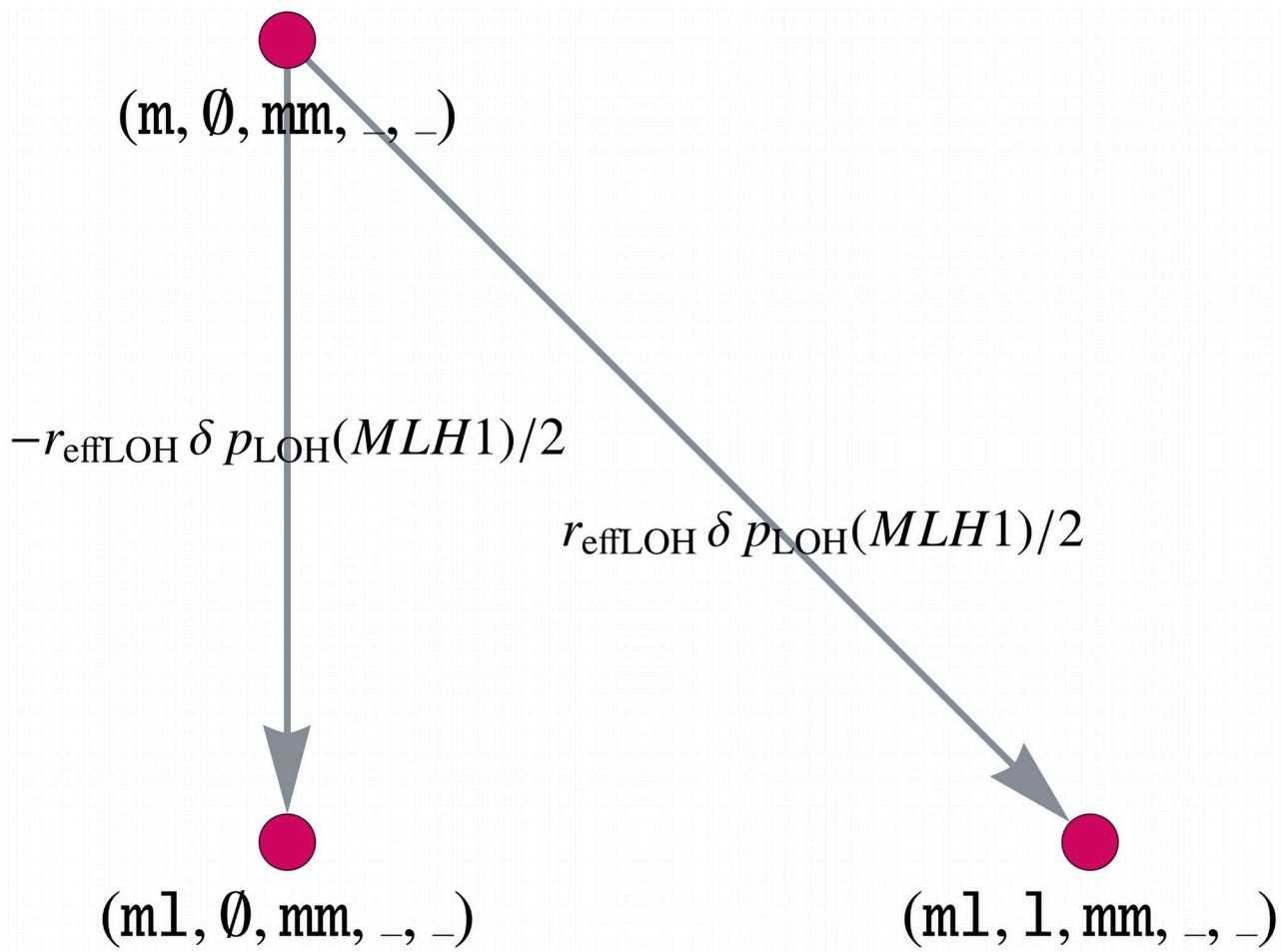

**Fig 7. Model component for the mutual enhancement of two dependencies by a factor $\delta r_{\text{effLOH}}$.** Part of the gene mutation graph for *CTNNB1* and *MLH1* after *APC* inactivation considered by the component *E*. The gene mutation graphs for the other possible gene states $MLH1 \in \{\texttt{l}, \texttt{ll}\}$, $CTNNB1 \in \{\texttt{m}, \texttt{ml}\}$, $APC \in \{\texttt{ml}\}$ are defined in an analogous way.

For this, the matrix *F* is defined analogously to the matrix *B* with the corresponding matrix entries multiplied by $\zeta$. The gene mutation graph of *KRAS* is given in Fig 8.

## 3.2 Modifications to the model

In Section 3.1, we introduced a mathematical modeling approach for colorectal carcinogenesis using the example of Lynch syndrome. We will present modifications to the model to handle other forms of colorectal carcinogenesis such as Lynch-like and MSS carcinogenesis, as well as colorectal carcinogenesis in FAP individuals.

For example, this can be done by changing the initial values of the model to differentiate between sporadic and hereditary cases or to consider germline variants in different genes, e.g., MMR in Lynch syndrome and *APC* in FAP.

Further, we can include other mutation status of already included genes, for instance the wild-type state in the MMR gene for the Lynch-like and sporadic MSI case, and we can adapt specific parameters to account for specific carcinogenesis mechanisms like we will do for the example of FAP later in this section.

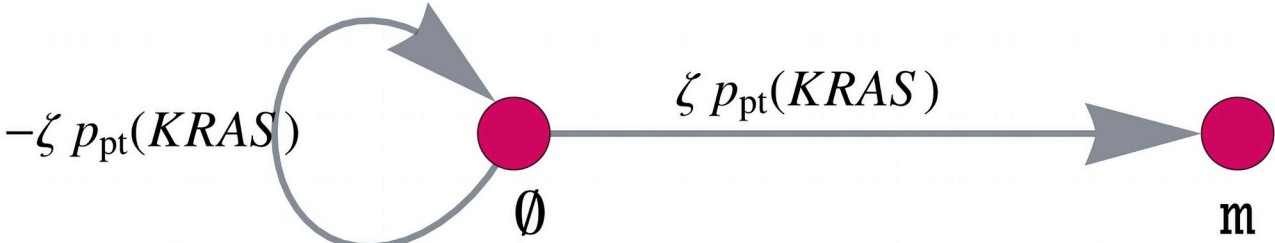

**Fig 8. Model component for increasing the mutation rate of *KRAS* after MMR deficiency.** Gene mutation graph of *KRAS* for the matrix *F* with the *KRAS* mutation rate increased by a factor $\zeta$.

Finally, we describe the potential for modifications to account for cancer evolution in other organs.

**3.2.1 Non-Lynch and FAP.   Lynch-like and Lynch syndrome carcinogenesis.** The main difference between Lynch-like and Lynch syndrome carcinogenesis is the absence or presence of a monoallelic MMR germline variant as a first hit at birth. In Lynch syndrome carcinogenesis, all body cells, including those constituting colonic crypts, already carry a monoallelic variant in one of the MMR genes, whereas in Lynch-like carcinogenesis all cells start with wild-type MMR genes. By introducing the additional vertex $\emptyset$ in $\mathcal{V}_{\mathrm{MMR}} = \{\emptyset, \mathrm{m}, \mathrm{l}, \mathrm{mm}, \mathrm{ml}, \mathrm{ll}\}$ with point mutation and LOH rates described in Sections 3.1.2 and 3.1.3, it is possible to represent those two forms of MSI carcinogenesis. The initial value changes to $x_0 = 0$ except for the entry corresponding to $(\mathrm{m}, \emptyset, \emptyset, \emptyset, \emptyset)$ or $(\mathrm{l}, \emptyset, \emptyset, \emptyset, \emptyset)$ in the hereditary case and $(\emptyset, \emptyset, \emptyset, \emptyset, \emptyset, \emptyset)$ in the sporadic case for which the value is set to $n_{\mathrm{crypts}}$.

**MSS carcinogenesis.** It is possible to model the evolution of MSS CRCs with the proposed model by not including MMR genes in the vertex set. Due to the absence of MMR in the model, *CTNNB1* mutations are much less frequent. The classical adenoma-carcinoma model including *APC*, *KRAS* and *TP53* is the dominant pathway of carcinogenesis.

**FAP carcinogenesis.** Another application of the model is the evolution of CRCs in another hereditary syndrome, namely FAP. Those individuals have a single germline variant in *APC*, which is known to be a point mutation in almost all cases [75, 76]. Thus, the dynamical system starts with all crypts in the state $(\emptyset, \emptyset, \mathrm{m}, \emptyset, \emptyset)$.

As reported in [77], we assume that the germline variants are not equally distributed among the base pairs of the *APC* gene. Instead, they are concentrated at specific codons leading to the fact that we change the number of hotspot base pairs in the FAP case. Due to [78], the classical FAP case is associated with germline variants in codons $1250-1464$, leading to the assumption $n_{\mathrm{hs}} = 600$ in our model for FAP simulations. Thus by changing the parameters of the model, we are able to model other cases of colorectal carcinogenesis.

The common regions of germline variants described above are also correlated with the most occurring polyps (more than 5,000) [78] in FAP individuals. With an estimated diameter of 4.8 mm per polyp [79] and 0.09 mm per crypt [80], this would result in $10^7$ crypts in a polypous state. Thus, our model simulations should also reflect that the number of polyps, assumed to consist of *APC-/-* crypts, should be much higher than in the sporadic case.

**3.2.2 Cancer in other organs.**   In general, it is possible to modify the model in such a way that it can not only model carcinogenesis in the colon but also in other organs. For this, the incorporated genes have to be changed as well as the definitions of point mutations and LOH events have to be adapted to account for different cell structures. The application to other organs will be considered in future work.

## 4 Results

We present the results of modeling the evolution of human colorectal crypts in a typical Lynch syndrome patient over the course of 70 years. The model starts with a germline variant in MMR in all crypts at birth and yields the temporal evolution of the crypt distribution among all genotypic states, where we only show the results for *MLH1* and *MSH2*, as those are related to the highest CRC incidence in Lynch syndrome [25].

### 4.1 Evolution of crypts with specific genotypic states

Making use of Eq (S1–15) in S1 Appendix, we extracted and combined different genotypic states from the overall distribution. We did so for MMR-deficient crypts as well as other more advanced states, which we refer to adenomatous and cancerous states. They are defined in the following way:

**MMR-deficient:** MMR-deficient; *CTNNB1*, *APC*, *KRAS*, *TP53* intact, i.e., $(\text{mm}, \emptyset, \emptyset, \emptyset, \emptyset)$ + $(\text{ml}, \emptyset, \emptyset, \emptyset, \emptyset)$ + $(\text{ll}, \emptyset, \emptyset, \emptyset, \emptyset)$

**State 1:** MMR-proficient or MMR-deficient, *CTNNB1* activated; *APC* inactivated; *KRAS* and *TP53* intact (called early adenomatous)

**State 2:** MMR-proficient or MMR-deficient, *CTNNB1* activated; *APC* inactivated; *KRAS* activated; *TP53* intact (called late adenomatous)

**State 3:** MMR-proficient or MMR-deficient, *CTNNB1* activated; *APC* and *TP53* inactivated; *KRAS* activated (called cancerous)

The parameters are set in such a way that the number of MMR-deficient crypts is quantitatively comparable to the clinical data presented in [80]. We show the results for *MLH1* and *MSH2* in Fig 9. The impact of the parameters on the simulation results are discussed in Section 4.4. The procedures for parameter learning and sensitivity analysis are planned to be included in a more mathematically focused follow-up work.

Further, the results for early and advanced adenomatous and cancerous states are given in Fig 10 for a typical Lynch syndrome patient with a germline variant in *MLH1*. It is important to note that we can analyze, e.g., the relative contribution of MMR-deficient and MMR-proficient adenomatous and cancerous states. With the chosen parameter combinations, this relative contribution changes between the advanced adenomatous and the cancerous states. We will further elaborate these contributions in Section 4.3. Further, it is possible to compare the evolution of these states with respect to the contribution of *APC* and *CTNNB1*. Note that some of the parameters are chosen without any bio-molecular data at hand meaning that some of the absolute numbers of crypts presented here may not match the real numbers once measurable. With increasing data available for the mutation rates or the evolution of crypt numbers, the model parameters can be adapted to further improve the similarity of the model output to clinical observations.

### 4.2 Influences of variants in MMR genes

The model is able to compare the carcinogenesis process for the different MMR genes in order to examine gene-specific differences. This in particular includes the questions of whether and how the distribution of crypts in various states changes when considering different MMR genes. More generally, the distribution among the different pathways of Lynch syndrome carcinogenesis may vary among the MMR genes. As the different pathways of carcinogenesis need different treatment and surveillance strategies, it is essential for Lynch syndrome-related

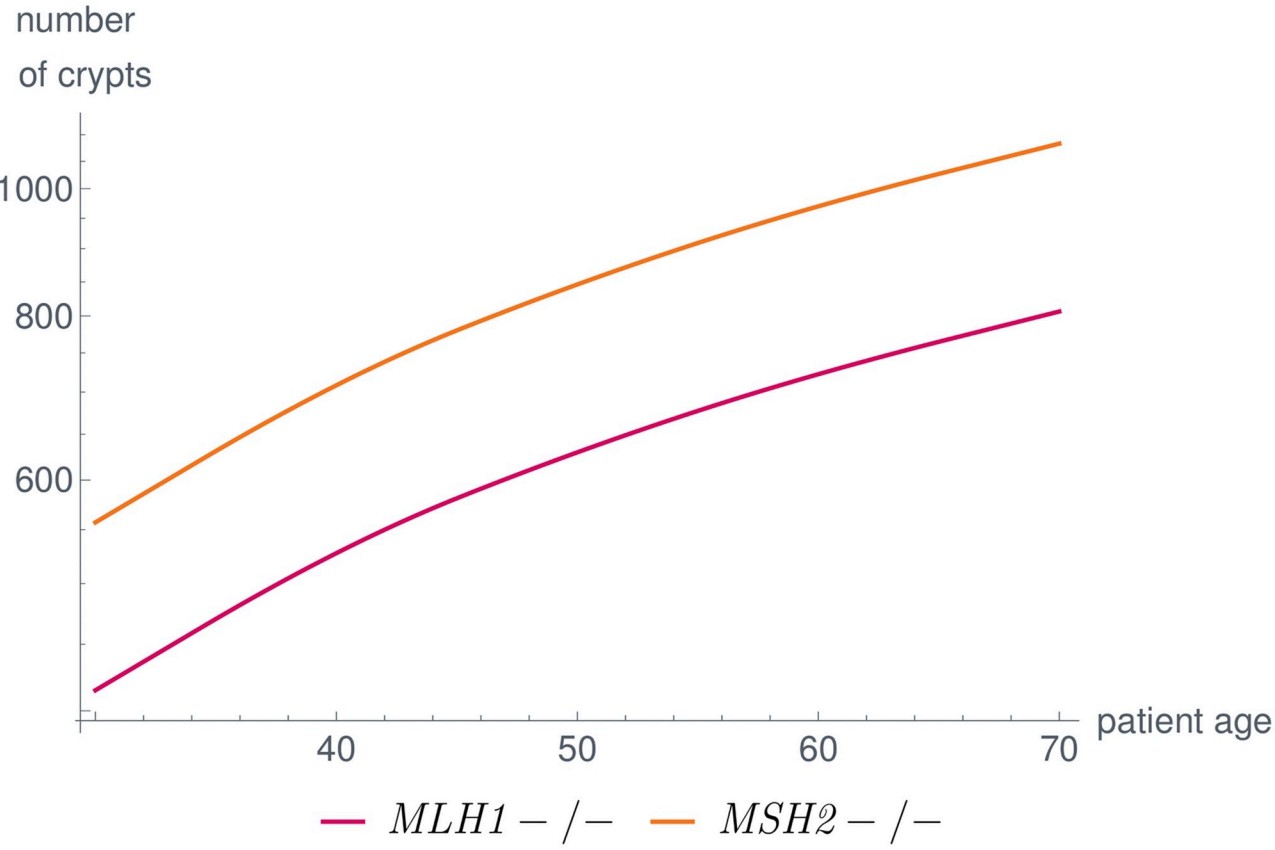

**Fig 9. Number of MMR-deficient crypts over the life of a typical Lynch syndrome patient for *MLH1* and *MSH2*.** The parameters in the model are set in such a way that the simulation results are in concordance with published data [80]. In our model, differences among genes are due to differences in coding region and gene lengths as well as the magnitude of the effects of the dependent mutational processes.

clinical guidelines to examine the gene-specific associations with the pathways of carcinogenesis, as depicted in [25].

An early example is given in Fig 9 showing the differences among MMR-deficient crypt foci which are the first detectable precursor lesions of the Lynch syndrome carcinogenesis pathways 2 and 3 illustrated in Fig 1. Differences among the MMR genes are reported for

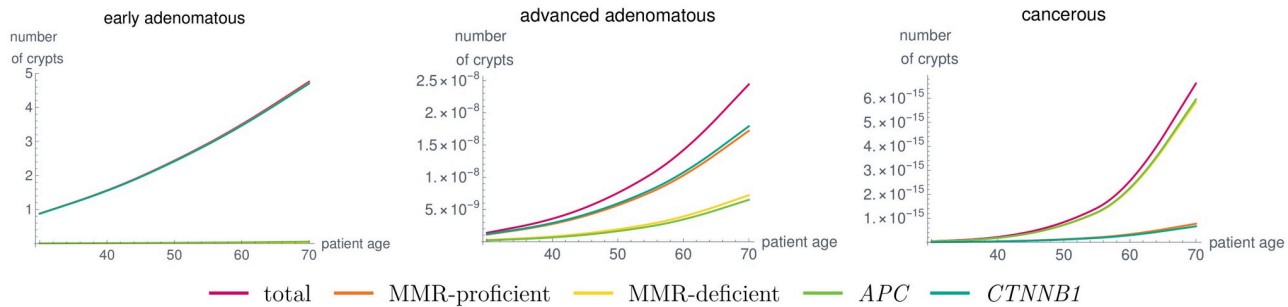

**Fig 10. Number of crypts over time in a typical *MLH1* carrier in combined states, like early adenomatous, advanced adenomatous and cancerous states as defined in the text for the given parameter set.** Due to the model components accounting for different genetic dependencies, the distribution of MMR-deficient and MMR-proficient, as well as the contribution of *APC* and *CTNNB1* change for the different states. Due to the lack of suitable medical data, parameter learning was not performed in a rigorous way. As soon as data are available, this can be done using different mathematical techniques.

adenoma and carcinoma incidences of Lynch syndrome individuals [25]. In the model, the differences are due to differences in the properties of the MMR genes, such as coding region and gene lengths, and due to the fact that dependent mutational processes influence the evolution of the crypts differently. As soon as there are more data available on bio-molecular mechanisms or there are further pathogenic variant hypotheses to be tested, these differences can be made even more explicit by introducing additional model components. This will be the subject of future work.

### 4.3 Distribution among the carcinogenesis pathways

We analyzed the proportion of MMR-proficient and MMR-deficient crypts in various states to determine the proportion in which MMR deficiency occurred as an initial event in carcinogenesis of Lynch syndrome carriers. The results are shown in Fig 11 and are similar to the currently available data [22] with a slight underestimation of MMR-deficient *APC-/-* crypts compared to MMR-proficient ones.

In general, for independent mutational processes, the distributions in Fig 11 are the same as there are no influences between the different genes. In our model, we can recognize the dependencies, as the distributions vary within the subsequent states. From *APC-/-* to *APC-/-* and *KRAS*-activated crypts, the difference in the proportions of MMR-proficient and MMR-deficient crypts greatly increases with the given parameter setting leading to the fact that almost all *APC-/-*, *KRAS*-activated crypts are MMR-deficient. As more of the *APC-/-* crypts are MMR-deficient, this seems to imply that MMR deficiency is often the initial event in Lynch syndrome carcinogenesis.

Further, the proportions do not change if *TP53* inactivation happens because currently, there is no such effect incorporated in our model for, e.g., increasing the mutation rate of *TP53* after MMR deficiency or after *KRAS* activation.

### 4.4 Analysis of parameter contributions

The results were obtained with the set of parameters given in Table 3. We analyzed the influences of the parameters on the simulation results. First, the number of point mutations $n_{pt}$, the number of cells $n_{cells}$, and the number of crypts $n_{crypts}$ determine the absolute values of the analyzed numbers.

Further, the relation of the hotspot length and the gene length determines the relative frequency of point mutations and LOH events for the individual genes, which can be changed by

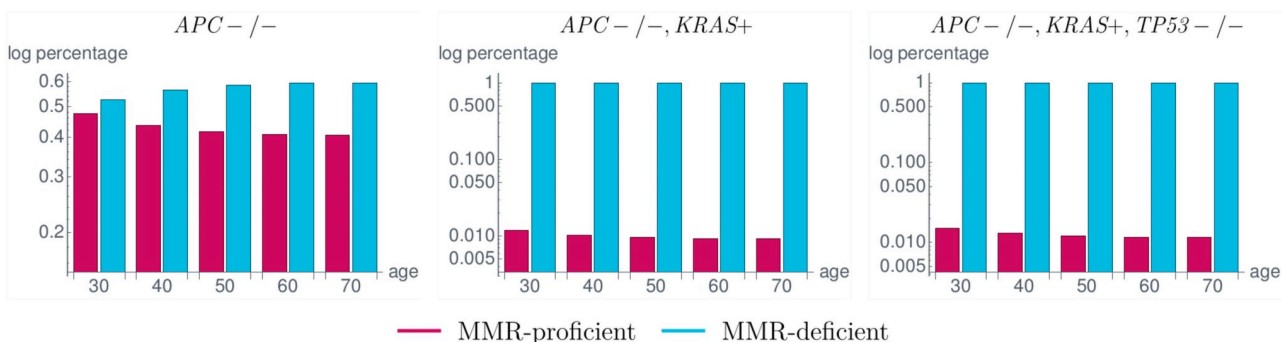

**Fig 11. Proportion of MMR-proficient and MMR-deficient crypts in a typical *MLH1* carrier in different states corresponding to the states in the classical adenoma-carcinoma sequence by Vogelstein [7].** Among the *APC-/-* crypts (left), the number of MMR-deficient crypts is up to 20% higher than the number of MMR-proficient ones. This difference largely increases with the subsequent *KRAS* activation (*KRAS+*) (middle) and *TP53* inactivation (*TP53-/-*) (right) leading to the fact that almost all crypts in the last state, corresponding to a cancerous state, are MMR-deficient. These simulation results are in concordance with available data with a slight underestimation of MMR-deficient *APC-/-* crypts [22].

  

**Table 3. Parameter setting for the shown results.**

| Parameter | Value |
|---|---|
| $n_{\text{crypts}}$ | $9.95 \cdot 10^6$ |
| $n_{\text{cells}}$ | $1.5 \cdot 10^3$ |
| $n_{\text{bp,genome}}$ | $3.2 \cdot 10^9$ |
| $n_{\text{pt}}$ | $1.2$ |
| $b(MMR)$ | $-0.01$ |
| $b(CTNNB1)$ | $0.0$ |
| $b(APC)$ | $0.10$ |
| $b(KRAS)$ | $0.01$ |
| $b(TP53)$ | $0.0$ |
| $f(MMR)$ | $2.3 \cdot 10^{-6}$ |
| $f(CTNNB1)$ | $1.2 \cdot 10^{-3}$ |
| $f(APC)$ | $8.3 \cdot 10^{-7}$ |
| $f(KRAS)$ | $2.5 \cdot 10^{-8}$ |
| $f(TP53)$ | $1.2 \cdot 10^{-5}$ |
| $r_{\text{effLOH}}$ | $0.9$ |
| $\beta$ | $10^3$ |
| $\delta$ | $10^2$ |
| $\zeta$ | $10^2$ |

including mutational dependencies for specific genotypic states. Here, the magnitude of the parameters $r_{\text{effLOH}}$, $\beta$, $\delta$, and $\zeta$ determines how large the contribution of the individual dependency is.

The parameters $b(\text{gene})$ affect the slope of the crypt evolution curve. In our case, $b(MMR) < 0$ leads to the fact that further MMR-deficient crypts are disadvantageous for the crypt survival leading to fewer additional MMR-deficient crypts with increasing age (Fig 9).

In contrast, *APC* inactivation is modeled as an advantage for the crypts such that $b(APC) > 0$ leads to more additional *APC*-inactivated crypts with increasing age.

Furthermore, the relation of the fixation affinities $f(\text{gene})$ for different genes seems to influence the ordering of the mutations. A larger value of $f(\text{gene})$ leads to a faster fixation in this gene and thus to an earlier event in carcinogenesis (Fig 11).

However, there is still uncertainty in the data about the fitness advantages and disadvantages of individual genetic changes as well as on the fixation affinities of mutations. General information on mutational dependencies and how they affect the phenotype of the cells is crucial to include further bio-molecular mechanisms.

## 4.5 Non-Lynch and FAP

We compared different types of colorectal carcinogenesis by changing the initial values of the dynamical system or by adapting other parameters.

First, we compared the number of MMR-deficient crypts in Lynch-like and Lynch syndrome individuals, as illustrated in Fig 12. The latter is much larger in Lynch syndrome individuals than in Lynch-like individuals, corresponding with [80].

This is due to the fact that in Lynch syndrome, a germline variant in one allele of the MMR gene is already present such that an additional somatic mutation leading to MMR-deficiency could be gained earlier in life.

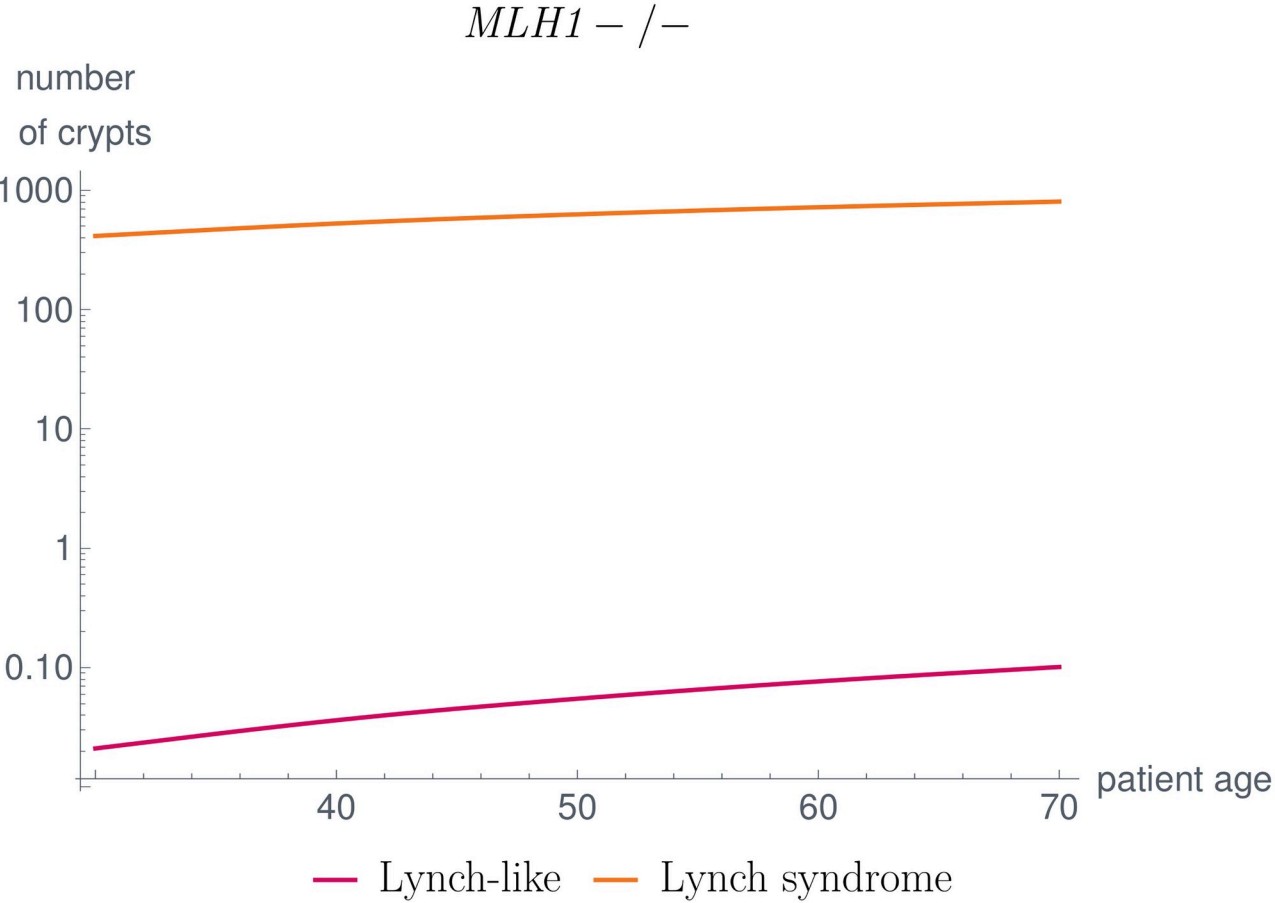

**Fig 12. Comparison of MMR-deficient crypts in Lynch-like and Lynch syndrome individuals.** The number of MMR-deficient crypts is significantly higher in Lynch syndrome individuals compared to Lynch-like individuals, which matches the findings in [80].

Further, we compared the *APC-/-* crypt evolution of a typical FAP patient with a sporadic case without a germline variant in *APC* for all crypts. We used the parameter setting given in Table 3, except for $n_{hs}(APC) = 600$. We changed the number of hotspot base pairs in the FAP case due to the fact that the germline variants are not equally distributed among the base pairs of the *APC* gene, as described in Section 3.2.1.

With the given parameter set, our model simulations yield between $10^4$–$10^5$ *APC-/-* crypts, which is below the estimates calculated from the literature (see Section 3.2.1). The time evolution of the number of crypts is shown in Fig 13. It would be necessary for the future to obtain age-dependent data as well as further measurements to be able to adapt the parameters accordingly.

## 5 Discussion

We presented a mathematical model for the multiple pathways of colorectal carcinogenesis based on a dynamical system with Kronecker structure, which models the number of colorectal crypts being present in different genotypic states.

The modeling approach consists of different model components for independent and dependent mutational processes. Although the Cancer Dependency Map [81] provides a great resource and extensive information about gene dependencies, data for specific medical

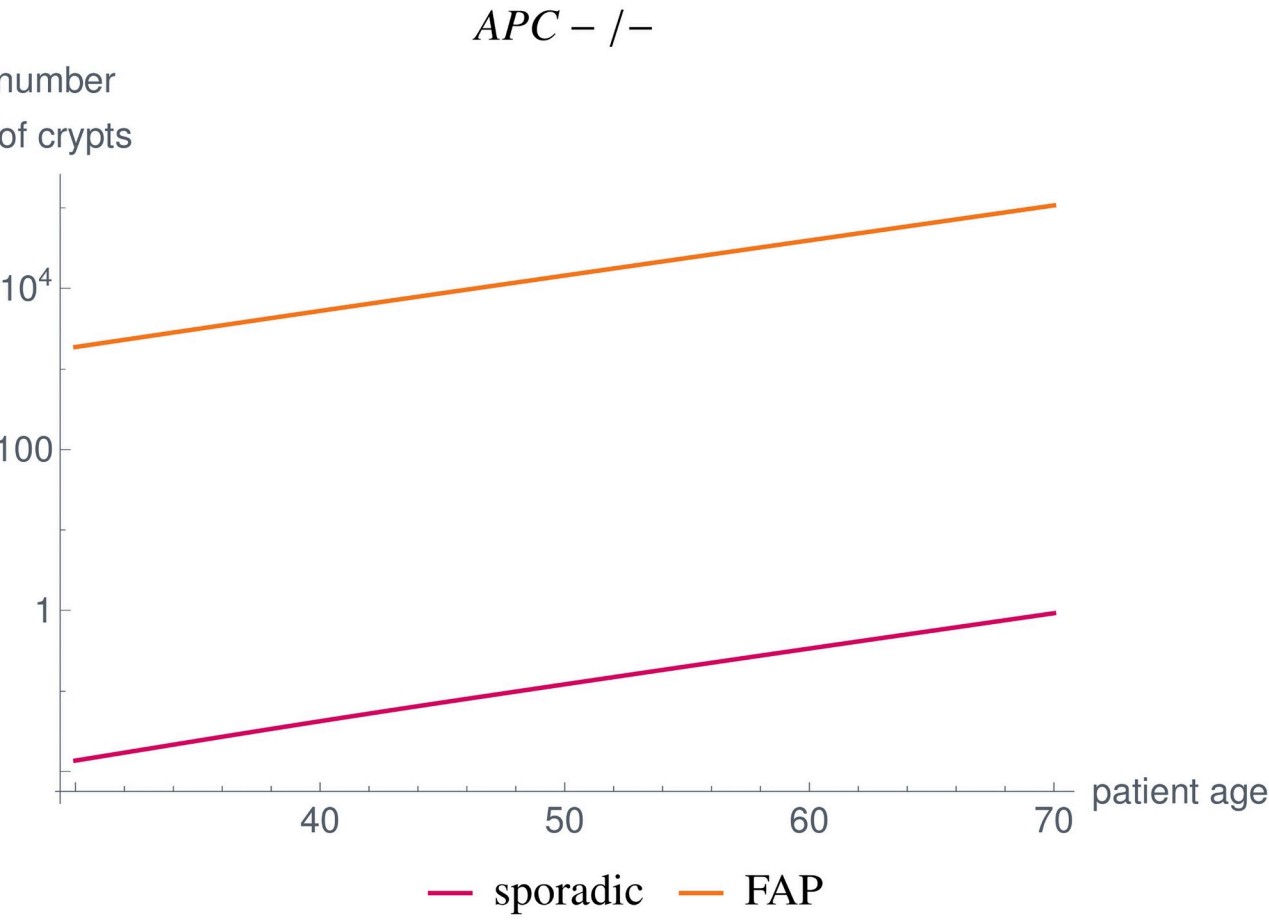

**Fig 13. Comparison of *APC*-/- crypts in the sporadic case and in FAP individuals, where we changed the initial value of the dynamical system as well as $n_{hs}(APC) = 600$ for FAP.** Our simulation results yield numbers below estimates found in the literature [78–80]. With improved measurements, future work will adapt the parameters accordingly.

contexts are scarce. Thus, the first model component is neutral and starts with the artificial assumption of complete independence. The process of adaptation to known dependencies is illustrated in our example of Lynch syndrome carcinogenesis.

Mathematically, the independence is represented by building mutation graphs for all genes individually and combining them using the Cartesian graph product. This means that the matrix of the corresponding model component can be obtained by combining the adjacency matrices using the Kronecker sum. The use of the Cartesian graph product is based on three assumptions: 1) the genotypic states in the combined graph are exactly the combination of the mutation status of the individual genes. This is a natural choice and not a limitation of the model. If there were additional genotypic states which should be considered, then they would be included in the individual genes already. 2) There is only one mutation at any point in time. However, simultaneous mutations can be included explicitly in the model. This is for example already done in the case of *MLH1* and *CTNNB1*. 3) The mutations considered in this model component are independent of each other. This is true for those mutations with data suggesting independence or due to lack of data indicating dependency. However, if there are data suggesting any dependency, this is considered in other model components.

The model includes further components representing specific correlations and dependencies of genetic events which are chosen in concordance with existing medical hypotheses and data. The corresponding matrices again have a Kronecker structure. Further, all matrices are combined in an additive way which eases the analysis of the individual effects on the overall model solution. In addition, if further medical hypotheses and data are available, it is straightforward to include further mutation dependencies in the model.

As an example, we focused on the evolution of key genotypic states occurring in Lynch syndrome, the most common inherited CRC syndrome, namely alterations in the MMR genes, with focus on *MLH1* and *MSH2*, *CTNNB1*, *APC*, *KRAS* and *TP53*. There might be other driver mutations in Lynch syndrome-associated colorectal carcinogenesis where empirical data are scarce and thus, these mutations are currently not covered for the specific example of Lynch syndrome modeling. Due to the general structure of the model, it would be possible to consider other driver mutations in future.

In order to apply the modeling approach to Lynch syndrome carcinogenesis, we assume gene-dependent mutation and LOH event rates meaning that the mutation rate of a gene is proportional to the length of the gene and the total number of mutations occurring in a cell during cell division. As there are multiple cells within a crypt each having an individual cell cycle, it takes some time until the mutation is present in the whole crypt, a process called fixation. Further, a mutation could be washed out of the crypt, if it is not advantageous enough for fixation to occur. Thus, we assume that the mutation rate of a gene in a crypt also depends on a fixation tendency of the specific genetic event. The edge weights in the graph representation correspond to the mutation rates between those genotypic states of crypts, where the mutation rates are computed based on the described assumptions.

By this choice of parameters, we were able to obtain simulation results which are in concordance with clinical observations. This includes the number of crypts in a specific genotypic state, like MMR-deficient crypts which are early precursors in Lynch syndrome carcinogenesis [80]. Further, we analyzed the influence of variants in different MMR genes, here for *MLH1* and *MSH2* as an example, leading to differences in numbers of crypts in specific states. This was recently observed in clinical data [25] suggesting adaptation of Lynch syndrome surveillance guidelines based on MMR gene variants. Here, rigorous analysis of the impact of MMR gene variants, considering also other MMR genes, and other molecular differences is subject of future work.

We are fully aware of the fact that our simulation results are depending on specific *a priori* assumptions. Moreover, our model is deterministic; therefore, options for assessment of robustness are limited and mainly based on parameter variations. Therefore, development of stochastic modeling approaches is desirable to more faithfully reflect natural cancer evolution, including random events and spontaneous disappearance of precancerous and potentially even cancerous lesions.

We analyzed the proportion of MMR-deficient and MMR-proficient crypts showing *APC* inactivation as a first indicator for the distribution among the three currently hypothesized pathways of carcinogenesis in Lynch syndrome individuals, with a good concordance to current clinical observations [22]. Future studies will include a more systematic analysis and modeling of this aspect.

The model can be easily modified to other types of carcinogenesis, such as sporadic MMR-deficient cancers, Lynch-like MMR-deficient cancers, other hereditary CRCs like FAP, and microsatellite-stable CRCs.

It is important to note that the modeling approach in general is independent of the specific parameter values. Thus, different assumptions for the mutation rates of individual genes can be used, if appropriate, for another carcinogenesis scenario. Moreover, different assumptions

for Lynch syndrome carcinogenesis, e.g., the inclusion of the `11` states or dominant-negative effects can be accounted for by adapting parameter values.

In principle, it is possible to apply the model structure to other organs by modifying the mutation probability definitions according to the underlying cell structure and by incorporating different genes with appropriate predominant genetic effects. This will be the subject of further investigation. Further, in the presented example, the model components are based on individual genes and gene-specific aspects. In other words, we consider genes individually and not their signaling pathways as entities. However, in general, it is possible to represent the model components by signaling pathways and the influence of alterations thereof.

In summary, we model carcinogenesis on the basis of the number of crypts being present with specific genotypic states. The latter can be aligned to clinically defined stages such as early adenoma, although we are fully aware of the fact that the congruence between clinical and molecular definitions will be limited due to the dynamics of cancer evolution and the limited availability of comprehensive data. Limitations of data also concern the topic of overdiagnosis and disappearing lesions. From a mathematical point of view, it is straightforward to include spontaneous disappearance of lesions in the modeling approach, as shown in the manuscript. However, there are currently not enough prospective data available to estimate or learn the necessary parameters, e.g., the probability of spontaneous crypt loss for each mutation status. This is the reason why we have chosen a simpler model jointly modeling the proliferation and disappearance by the self-loops in the graph, largely reducing the number of parameters that need to be determined. If more molecular data with the analysis of all possibly relevant genes are available, a comparison of the model with these data will allow for parameter learning of the yet unmeasurable parameters. In this context, we would like to emphasize that the "linear model" used in the present approach only reflects the mathematical framework of linear differential equations, but does not represent the evolutionary process, which we consider as a parallel, competitive process of mutational events, persistence and regression of lesions.

Further, the modular structure of the model allows for an inclusion of further states, e.g., death/disappearing states in a natural way. This also concerns external factors, such as effects of the microenvironment or the role of the immune system: Our model, through the flexibility regarding mutational events and their consequences, can also be used to make specific assumptions about tumor-immune cell interactions, for example assuming a higher immune visibility of MMR-deficient cell clones with high mutation load, which is part of future work.

## Supporting information

**S1 Appendix. Mathematical background.** This includes basic notions from graph theory, graph products, the Kronecker sum of matrices, linear dynamical systems and their solution. (PDF)

## Author Contributions

**Conceptualization:** Saskia Haupt, Alexander Zeilmann, Matthias Kloor, Vincent Heuveline.

**Data curation:** Saskia Haupt, Vincent Heuveline.

**Formal analysis:** Saskia Haupt, Alexander Zeilmann, Vincent Heuveline.

**Funding acquisition:** Magnus von Knebel Doeberitz, Matthias Kloor, Vincent Heuveline.

**Investigation:** Saskia Haupt, Alexander Zeilmann.

**Methodology:** Saskia Haupt, Alexander Zeilmann, Vincent Heuveline.

**Project administration:** Vincent Heuveline.

**Resources:** Aysel Ahadova, Hendrik Bläker, Magnus von Knebel Doeberitz, Matthias Kloor.

**Software:** Saskia Haupt, Vincent Heuveline.

**Supervision:** Magnus von Knebel Doeberitz, Matthias Kloor, Vincent Heuveline.

**Validation:** Saskia Haupt, Matthias Kloor.

**Visualization:** Saskia Haupt, Alexander Zeilmann.

**Writing – original draft:** Saskia Haupt, Alexander Zeilmann, Aysel Ahadova, Matthias Kloor, Vincent Heuveline.

**Writing – review & editing:** Saskia Haupt, Alexander Zeilmann, Aysel Ahadova, Hendrik Bläker, Magnus von Knebel Doeberitz, Matthias Kloor, Vincent Heuveline.

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
