## [Decision Letter · Decision Letter 0]

18 Nov 2020

Dear Mrs Haupt,

Thank you very much for submitting your manuscript "Mathematical Modeling of Multiple Pathways in Colorectal Carcinogenesis using Dynamical Systems with Kronecker Structure" for consideration at PLOS Computational Biology.

As with all papers reviewed by the journal, your manuscript was reviewed by members of the editorial board and by several independent reviewers. In light of the reviews (below this email), we would like to invite the resubmission of a significantly-revised version that takes into account the reviewers' comments.

We cannot make any decision about publication until we have seen the revised manuscript and your response to the reviewers' comments. Your revised manuscript is also likely to be sent to reviewers for further evaluation.

Sincerely,

Jing Chen

Guest Editor

PLOS Computational Biology

Natalia Komarova

Deputy Editor

PLOS Computational Biology

Reviewer's Responses to Questions

**Comments to the Authors:**

Reviewer #1: I below comment on the starting points/assumptions for the reasoning and leave the correctness of the specific mathematics and its references to a professional statistical expert.

The paper is very difficult to read, partly because lack of clear organization: Neither the definition of terms used, the basic conceptional understanding of what is ment by ‘cancer’ (and which seemingly is confused with ‘tumor’ in parts of the mns) and the basic assumptions underlying the statistics are not presented distinctly do the reader, and the obligatory section discussing the impact of these/violation of these I do not find. Deep inside the text I find three key factors mentioned:

1. the order in which mutations are accumulated is ignored

2. The edge connectivity assumption states that no two mutations in different genes can occur at the exactly same point in time

3. The third assumption entails that a mutation in one gene does not change the mutation probability in another gene.

the first of which seemingly in conflict with the basic understanding that the order of the events is of interest as mentioned initially, the second partly invalid, and the third obviously wrong: If an initial mutation increases proliferation rate, the probability for one of the cells carrying the mutation to have a second hit is to me obviously increased – which is what we teach the student when explaining carcinogenesis. And which is why the order of mutations is believed to be significant. I do see that there may be ways to read the paper not to find these arguments to be contradictory, but a scientific paper is not to be text to be read in good faith to be understood correctly.

The shifts back-and-to between considering cancer/tumor in general and MMR-deficient tumours specifically has to me the flavor of a paper explaining something to the general public with examples, and is not what is expected in a scientific paper. These shifts make it very difficult to control the relationships between assumptions/input to calculations and results.

The intro mention immune system counteracting cancer development, but does not refer to the growing knowledge of ‘overdiagnosis’ – established cancer may stop growing and/or disappear. By and large the mns seemingly consider cancer a result of a process having some probability to occur, but does not consider the common understanding that these biological events are much more frequent than the end result cancer indicates because there are opposing system removing the mutated cells – it’s briefly mentioned but not discussed to my reading as a major factor. When considering Lynch Syndrome, this is of importance because the initiating factor (MMR deficiency) may/will all along the way constantly impose new mutations which may both block the effect of first mutations and trigger new biological systems with the effect of Darwinistic reduced fitness of the growing cells/clone. This to me is an element of probability, and which seemingly is denied in the assumptions:

“Most of the cancers in the general population occur by chance. These cancers are called sporadic. However, in some families, certain types of cancer appear more frequently. This is either a familial or a hereditary form of cancer…. From a modeling point of view, the advantage of focusing on hereditary tumors is that there are clearly defined molecular events determining the onset of the disease and thereby representing a known mechanism underlying carcinogenesis.”

To this referee, these assumptions are questionable, I would rather say that any individual has a large set of inherent and acquired probabilities for biological mechanisms going wrong, and the difference between familial/inherited cancers and ‘sporadic’ ones is that the familial/inhered cancers cases have different probabilities for some unwanted outcomes in the billions of biological events happening in their cells each second they live. Following such a way of thinking, cancer is close to inevitable if one live long enough, but with increasingly good health and longliving the probability for contracting cancer increases because the incidence of dying before you so do decreases. In such a context, the probability for being live born and live for many years with a significant failure in one of the important system to mend errors, is about a small brim of errors between what is ‘normal’ and what is compatible with life. Boiling it down to a short analogy: The human body is not a machine, because it has the inherent capability of repairing itself, and to me this basic understanding is not good enough visible in the mns.

In sum, to me the mns was interesting reading, but the validity of the conclusions I cannot describe because the authors to me don’t.

Reviewer #2: This is a solid paper that describes a mathematical approach to deterministic modeling of cancer initiation. I have two concerns about the paper: 1) their model is extremely similar to the recently published Paterson et al. paper (Ref. 37) and 2) many of the parameter values are arbitrarily set, even when estimates are available from the literature.

Regarding issue 1), the authors mention the Paterson et al. paper, but are not clear as to how much of their model is virtually a verbatim copy of the Paterson et al. model. They consider 5 genes, APC, KRAS, TP53, CTNNB1 and "MMR", whereas Paterson et al. consider APC, KRAS and TP53. The graph of genotypes and the possible states for the three genes in common are exactly the same as in Paterson et al, as well as the general approach to calculating gene-specific numbers of driver positions, studying accumulation of mutations in a population of crypts (i.e. disregarding tissue hierarchy), assuming APC and KRAS provide fitness advantage but not TP53 etc. The Paterson et al. model is stochastic, but the authors use a dynamical system approximation to derive analytic results, which is also similar to the present paper. Even though the focus of the manuscript is on Lynch syndrome, the authors should clearly state that the approach they take is very similar to the one of Paterson et al. in many aspects.

My second issue is regarding how parameters were set.

For example, regarding results in Fig 10 the authors state: “Number of MMR-deficient Crypts over the Life of a typical Lynch Syndrome Patient for MLH1 and MSH2. The parameters in the model are set in such a way that the simulation results are in concordance with published data”. Why did the authors choose to set parameter values in this way? Since there are many parameters in the model, how did they choose these specific values? A better approach would have been to estimate all parameters from the literature, whenever possible. For example, b(KRAS) has been estimated to be 0.07/year (Ref. 67), and the authors set it to 0.01. Estimates similarly exist for b(APC) (Ref. 13).

There are also some formatting issues, e.g. eq. (1).

Reviewer #3: The manuscript entitled "Mathematical Modeling of Multiple Pathways in Colorectal Carcinogenesis using Dynamical Systems with Kronecker Structure" by Haupt and colleagues propose an extensible mathematical model of colorectal carcinogenesis for Lynch syndrome based on Kronecker sum of mutation graphs adjacency matrices. The authors focus on an important problem, however they hardly put their work into context. Modeling hereditary colorectal cancer in terms of genomic alterations occurring in defined pathways is interesting, however the enthusiasm for this work is highly diminished by a few major shortcomings, as discussed below.

Major comments:

1) it is not clear what purpose this whole exercise will serve. What does this model does for a patient? The authors mention vaguely "individual needs for prevention and treatment", but they do not make any effort to develop this point.

2) The mathematical modeling is very complicated, for sure inaccessible to the (lay) audience working in the cancer research field. The authors should make a substantial effort to make it less obscure.

3) The mathematical framework appears to be "ad hoc", with many assumptions that are not necessarily adequate, justifiable, or justified, so it is hard to appraise how generalizable this approach would be. Even the choice to focus on Lynch syndrome is partial, since this is not the "mainstream" path to carcinogenesis.

4) The paper is, in summary, a very long list of assumptions. Some of them questionable from the cancer biology point of view. Some sound, some very well explained, some not explained that well. An effort should be made justify all assumptions. That said, how generalizable is this approach if it depends on so many assumptions?

Below follows a few comments to support my review.

Specific comments

1) Abstract:

- The abstract requires major revision: after reading one can only guess what the manuscript is about. It appears it is about tumor evolution in a very specific hereditary syndrome, and that this should be important for developing more rational clinical treatments. Yet, this is not put into context, and the discussion of the mathematical models and their assumptions remains vaguely linked to the problem under investigation.

- The authors, in the abstract and introduction, make claims about "different types of colorectal cancer with individual needs for prevention and treatment." Yet these are not introduced and discussed. Explicitly grounding the work in current colorectal cancer research (and the related knowledge on molecular subtypes, etc...) would be highly beneficial.

- I definitely agree with the idea that "... a single model describing the whole process of carcinogenesis ... is too complex to build". I think that this issue is grounded in the reductionist (and often positivist) approach of most scientific research in this fiels (and others too, of course). And yes, "... the increasing medical knowledge ..." had made it more evident. Yet, even postulating the possibility (of a unifying theory) in the first place might be perhaps too ambitious. That said, the authors should perhaps consider to restructure the Introduction section to make it more consequential. Discussing mathematical modeling before discussing cancer and its pathogenesis is a little odd, and does not help the readerships to follow the narrative. More explicitly: sections 1.1 and 1.2, should go after Section 2, and maybe section 1.3 should come at the beginning.

- Citation 16 might not be the correct one: did the authors meant to cite: "doi: 10.1093/bioinformatics/btp505"?

- Lines 109-112: "..., while loss of heterozygosity (LOH) refers to the loss of larger regions of a chromosome, which can result in the deletion of whole genes." -> This is not correct, or well explained. LOH is not COPY NUMBER LOSS or COPY NUMBER ALTERATION... LOH is when one copy of a normally diploid genome is lost at some resolution (a focal loss, or a larger region, or a whole chromosome, it doesn't matter). This can result in loss of function, when coupled with mutations (or other alteration types) in the remaining copy. Please correct.

- Lines 113-119: Discarding 99% of the genome is "old school". Pathogenic, non-coding mutations affecting coding genes have been known for years (see the globin genes in thalassemia for instance) and the relevance of non coding genes is becoming more and more relevant. Hence, the following statement should be amended: "As about 99 % of the genome is not translated into proteins, most mutations do not have a direct consequence on cell viability or behavior."

- Lines 120-121: See above, therefore the following statement – "This means that, from all the possible mutations that can occur, we need to identify the relatively few key events which have a functional impact on the cell" – is not entirely correct and should be revised. There might be only a few CODING mutations that might be relevant, however we have no clues about the non-coding domain.

- Lines 123-125: There is also the possibility that a cancer is controlled by non-cell autonomous mechanisms, that are not defined by the combination of mutational events happening in the initiating cell (i.e., affecting oncogenes, tumor suppressors, and the detrimental mutations affecting the fitness of the cancer "to be" cell)... For instance, the role of immune surveillance... etc.

- Lines 126-128: "Different combinations of key mutations result in several distinct pathways." this sentence does not make sense. Signaling pathways exist independently from the mutations in the genes that encode the proteins participating in said pathways. Different combinations of mutations result in aberrations of specific pathways, usually non redundant ones. Furthermore, pathways don't "arise", they exist in a normal state, they dynamically regulates various processes in cell and tissue homeostasis, and are altered in response to genomic hits (whatever their nature is).

- Lines 128-131: The authors should put this "high-dimensional and complex problem" into context. Perhaps they should mention that a tumor harbors hundreds/thousands of mutations, and that they are scattered over many genes/pathways/processes, etc... Perhaps they should also define the "problem". Is the the problem making sense of all the possible mutation combinations, finding the one or few combinations of drivers? If yes, for a given cancer? For all cancers in a population? Since it is also mentioned "medical interpretability", are the authors referring to individual patients? If so, it should be explicitly stated and explained.

- Lines 134-136: "Mutations are errors which occur during DNA replication within cell division and which are not repaired by one of the error detection, repair and control systems present in all organisms." It is odd to explain the biochemistry of mutations in details here, after having discussed them before in the manuscript, giving for granted the reader was already well acquainted with the concept.

- Lines 147-152: perhaps when distinguishing hereditary from sporadic cancers the authors could introduce the concept of somatic versus germ-line mutation. The explanation provided is confusing, for instance using the word "genetic" ("The former is due to a combination of genetic and environmental factors...") points to the genetic material, the DNA, but it might also suggest heritability, etc... And so on... Making the distinction between "somatic" and "germ-line" mutation is key when discussing sporadic and hereditary cancers... Especially since somatic mutations are mentioned later on in the paragraph (line 158).

- Lines 168-173: Here it should perhaps be discussed the existence, in the crypts, of stem cells, and then mention tissue renewal, and why it is important to look at stem cells and hence crypts...

- Lines 180-185: here the authors seem to imply that MMR genes are not to be considered tumor suppressors... If they are not tumor suppressors, what are they? Aren't tumor suppressor genes involved, broadly, in these following processes?

1) Stop cells from dividing; 2) Repair DNA damage; and 3) Start programmed cell death.

So, it appears that MMR are bona-fide tumor suppressors...

B) Introduction – Section 3.

- Lines 215-217: "... the accumulation of mutations in two different genes of the same cell, ...", perhaps the authors want to say the "OCCURRENCE of mutations", since rarely in the same cell mutations accumulate in the same gene...

- Lines 230-237: about the "Independence of the Processes" it is stated that it "... entails that a mutation in one gene does not change the mutation probability in another gene." Is this assumption reasonable? MMR mutations are exactly the mutations that makes other mutations more likely. Similarly, mutations in other tumor suppressor genes are linked to subsequent genomic alterations, therefore this assumption is not correct. The possibility that this could be useful to create a "baseline" to measure how much this assumption is violated is besides the point... What value there is in measuring the deviation from a "wrong" baseline? The model derived can still be god at describing tumorigenesis. At least the authors could make the argument that this wrong assumption is necessary to start working, then... maybe down the road they will be able to revisit the whole thing and make better assumptions...

C) Section 4.1

- Lines 313-316: When the authors discuss the "ll" state, they mention that "... in CTNNB1, APC and TP53 damage a cell in such a way that it directly leads to cell death. Thus, there will be no crypt with all cells being in that state. As we model the evolution of genotypic states of crypts, we do not consider the ll states for CTNNB1, APC and TP53." It is not clear whether this assumption is reasonable. Let's take into consideration TP53. Usually mutations in this genes are dominant negative, so one hit is enough to have a phenotype, so to speak (despite TP53 being considered a tumor suppressor). So, parsimony in tumor evolution might explain better the absence of the "ll" state for this gene, rather then lethality... Furthermore, it is not clear why to distinguish "ll" from "mm"... Also two hits of "m" types can potentially be lethal, so it is not clear why to make a special case of "ll" and not "mm", or "ml" for these genes... This assumption might be justifiable, in some ways, even it could be grounded in tumor biology, but the authors should make a better case for it than they do. They could look into large genomic studies, and for instance have a look at the frequencies of double hits in their favourite genes, and perhaps empirically justify their assumption... It is true that TP53 is very rarely mutated twice, etc...

- Lines 319-322: these lines makes the assumption above even more confusing: the authors knows that TP53 is consider a tumor suppressor, they also know about dominant negative effect, yet... While they ignore this effect for APC, they make no comment about TP53... In other words, without some clear consequential explanations, it might look like the authors "bend" cancer biology as it is necessary.

Additional minor comments:

1) Abstract: "Like many other tumors, colorectal cancers develop through multiple pathways and different driver mutations." -> I would rephrased this stating that "... colorectal cancers through ALTERATIONS AFFECTING multiple pathways RESULTING FROM different driver mutations".

2) Abstract: "e suggest a linear autonomous dynamical system modeling the evolution of the different pathways of colorectal carcinogenesis." -> Unclear, please rephrase.

3) Figure one would be better explained by labeling the panels and then explaining them accordingly in the corresponding legend.

4) Introduction: references for the Lynch syndrome, MSI, etc should be added.

5) Please check format of references 9 and 10.

**Have all data underlying the figures and results presented in the manuscript been provided?**

Reviewer #1: Yes

Reviewer #2: Yes

Reviewer #3: Yes

PLOS authors have the option to publish the peer review history of their article (what does this mean?). If published, this will include your full peer review and any attached files.

Reviewer #1: **Yes: **Pål Møller

Reviewer #2: No

Reviewer #3: No
---

## [Decision Letter · Decision Letter 1]

21 Feb 2021

Dear Mrs Haupt,

Thank you very much for submitting your manuscript "Mathematical Modeling of Multiple Pathways in Colorectal Carcinogenesis using Dynamical Systems with Kronecker Structure" for consideration at PLOS Computational Biology. As with all papers reviewed by the journal, your manuscript was reviewed by members of the editorial board and by several independent reviewers. The reviewers appreciated the attention to an important topic. Based on the reviews, we are likely to accept this manuscript for publication, providing that you modify the manuscript according to the review recommendations, especially those about more accurate description of the background and assumptions, and discussions about the limitations of the work.

Sincerely,

Jing Chen

Guest Editor

PLOS Computational Biology

Natalia Komarova

Deputy Editor

PLOS Computational Biology

[LINK]

Reviewer's Responses to Questions

**Comments to the Authors:**

Reviewer #1: Firstly, I am grateful for the opportunity to discuss this paper with the authors, and trust them to understand that my expected contribution is to discuss problems and neither to give global statements nor promote parts to which I may agree. I also take the opportunity to add that the problems the authors address are well known to me, as are the difficulties to describe them in mathematical models which are as simple as possible (Descartes) but not too simple (Einstein). When IT systems now have the ability to undertake the numeric complexity of the analyses described, time is due to do so. When no one has all the knowledge needed to do so alone, each of us will have different focus. Mine are the assumptions used in the mathematical algorithms and their consequences.

In general, the authors have responded to my initial comments, but the mns is still very complex with many words and I still find problems needed to be clarified. I discuss these below partly in the order spotted when reading the revised mns and partly trying to make a readable narrative, not in the order of importance as is usual – because it is unclear to me to which degree they may modify the results, and the identification of the problems may be the only step to currently agree upon, and which in itself is the requirement to at some time to arrive at a conclusion.

If I had a solution to the problems I describe below, I would have submitted a paper with these myself, but I have not. In part the solution may be there is no solution, to which I may refer interested to the conflict between the works/conceptions by Newton and Hawkins on causation and probabilities (to put it simply: a rare stochastic event does not have a cause). My intention with describing the problems below, is to ask for a more clear definition of the terms used, the assumptions used, and a more distinct discussion of the limitations of the conclusions arrived at.

The definition of ‘cancer’ is still lacking, and combined with the terminology ‘a linear model’ and ‘cause of death’ one get the understanding that an untreated cancer will cause death. In contrast, the many crypts in the gut are constantly proliferating and the cells are expelled into the lumen and die. This may be the fate of MMR deficient crypt cells – growing precancers but not visible tumours because they are – in contrast to adenomas - constantly loosing their mass. Also, the immune system is known to recognize MMR deficient cells as ‘foreign’ and trying to kill them (cfr immunocompetent cell infiltrations, HLA associations, and vaccination attempts). The assumption (Vogelstein) in which we all believed some decades ago, was that 1) there is a limited number of precancerous tumour stage and we may prevent cancer if these are removed, 2) the initial cancer is local without spread and 3) if a manifest cancer is removed before spread the patient is cured/will not die. The initial observed effect of intervention (colonoscopy) did not meet the first assumption, and which was commonly agreed as the basis for shortening the intervals between colonoscopies. To our surprise, shorter intervals between colonoscopies had no effect on – or possibly increased – colon cancer incidence in LS which is incompatible with the first assumption. And – even more surprisingly – shorter intervals between colonoscopy has not been demonstrated to reduce mortality from colon cancer, which is incompatible with the second assumption. While I do agree that there is no knowledge indicating exactly how to quantify these problems (in sum referred to as ‘overdiagnosis’), it to me should be possible to estimate some parameters to make some examples (often referred to as iteration using statistical models to arrive at best fit with observations made). I do acknowledge that there are too many parameters and too heavy computational procedures to undertake Monte-Carlo-like simulations to search for strata testing all possible combinations and this is basically why the statistical models used have been developed to arrive at most probable overall results, but assuming some (extreme) parameters as a sort of sensitivity analysis to assess robustness of the results arrived at should be doable. When not done, to me, the results are ‘assuming this the results are that, but we don’t know to which degree the assumptions may be true and to which degree they may influence the results’.

As an illustration of the above, the leftmost diagram in Fig 1 is not what Ahadova et al published, but in contrast to their attempt to quantify the results, the figure is indicating that the bottom pathway starting with a MMR deficient crypt is dominating. Which is compliant with results in this paper, and should lead to a re-consideration of the assumptions made (iterating). The inherent problem of probabilistic modelling, is that each argument initially (a-priori) are to be given a value (NULL is not tolerated), and the values are not to be too low (zero a-priori value will give zero as results whatever the other arguments are when calculated as a product), and a very low a-priori value will be a probability ‘dark well’ from which one never may escape. To escape from this, a-priori values are often arbitrarily given as 0.5 and no arguments set to lower than 0.1 (cfr InSiGHT classification system for determining pathogenicity of the variants discussed in this paper). The resulting probabilities are in consequence by definition ‘false’ – they are not ‘real’ probabilities but the consequences of the assumptions given as a requirement to use the algorithms. While the independent mutations assumed in this mns have estimatable prevalences, the complex mutations I discuss have so low a priori probabilities that they may not easily be included in probability algorithms. If these actually are the causes of cancers killing the patients, however, the main results of this paper will not be valid to explain why some patients die and others do not. Again, this is about what is meant by ‘cancer’ and to which degree ‘cancer’ may be a surrogate for ‘death’. What I ask for is not an expansion in word to describe the principles in more details, but short statement(s) of results when including different a-priori assumptions in the calculations.

The statement “Mutations can either activate oncogenes (called gain-of-function mutations), which normally promote appropriate cell growth and proliferation, or mutations can damage or destroy tumor suppressor genes (called loss-of-function mutations), which normally limit cell growth and proliferation.” is misleading/not true: The consented view is that activation of oncogene transcription is through deactivating the normal homeostatic way these are controlled by suppressor genes through deranging the suppressor genes to loose their ability to suppress oncogene transcription. The oncogenes were denoted oncogenes by the misconception that mutations in these could increase their transcription, while the opposite is true: oncogenes loose their ability to increase cell division if mutated. To the end that mutations in suppressor genes are the main cause of gain of function of oncogenes. The normal homeostatic mechanisms activating oncogenes are intricate, and to the degree these are implicated in carcinogenesis it may be related to the inflammatory mechanisms associated to increased BMI now increasing the incidence of colorectal and endometrial cancer and which may be partly counteracted by COX inhibitors (aspirin and naproxene). Chronic myeloid leukemia, which is not inherited, is caused by a chimeric de-novo gene caused by a somatic mutation upstream to the functional coding part deranging the genes ability to be switched off by the normally controlling suppressor-genes. Such mutations increasing the transcription of an oncogene are by and large not compatible with life and not seen in newborns. They should not be mentioned in the current context. Whatever references one may have to the opposite, the old concepts I was told in medical school were misconceptions blocking understanding both normal homeostasis and carcinogenesis. The take-home message for true gain-of-function mutations should be that they are incompatible with life and therefore not inherited. (A more detailed discussion including inherited EPCAM tail deletion causing methylation/epigenetic inactivation of MSH2 having the same effect as a deranging variant in the MSH2 coding structure is outside the level of this comment). Inherited cancer is an observed (slightly) reduced capability of de-regulating oncogenes, leading to a slightly increased probability for carcinogenesis by age. In particular, dominantly inherited cancer is incompatible with substantially increased cancer incidence in fertile ages. Which is exemplified, but not discussed, when FAP is used as an example: the probability for a single adenoma caused by the CIN pathway to become a cancer is very low, and the disease is recessively inherited because heterozygotes have very low probability to produce null-allelic cells to become adenomas which in turn have a very low probability each to become cancers. The mixture of discussing FAP and LS together, the former a recessively inherited disorder where the adenoma rarely progress to cancer and where cancer by and large emerges in the adenomas, and LS where adenomas were assumed to be rare but obligate precursers with very high risk to become cancers, has been a major confounder prohibiting the understanding of LS carcinogenesis. The current mns to me repeat this confusion, while on the contrary it should distinctly be described as a misunderstanding.

The statements :” Lynch syndrome is caused by an inherited mismatch repair (MMR) gene mutation [4]. Colorectal cancers which develop due to Lynch syndrome therefore are MMR-deficient and show microsatellite instability (MSI) [5].” are not correct and misleading: As mentioned earlier, an inherited variant should not be denoted ‘mutation’, and the logical connection between the inherited variant causing cancer and the MSI phenotype is an observed association, not necessarily a logical consequence. The word ‘caused’ is in the present context of probability calculations misused: The inherited variant is associated with an increased probability of developing a cancer, and – actually - acquired mutations in the MLH1 genes may more frequently than inherited pathogenic variants cause MSI cancers. The major cause of an MSI cancer is not an ‘inherited’ variant. The concept of causation is theoretically very difficult and should be avoided in the current context: no single factor is ‘the cause’ of cancer – it is a coincidence of many factors.

This is again implicated in the following statement: “ The step-wise hypothesis has been validated subsequently in many independent studies for many different cancer types. Currently, it is expected that a minimum number of three mutation events is required to transform a normal cell into a cancer cell. This hypothesis is called the three strikes hypothesis [13].

Individuals with Lynch syndrome are predisposed to developing certain malignancies with a substantially higher lifetime risk compared to the general population. The most common Lynch syndrome manifestations are colorectal cancer (CRC, 50 % [19] compared to 6 % in the normal population) and endometrial cancer (40–60 % compared 128 to 2.6 % in women without Lynch syndrome) [4, 20]. Further, individuals have an increased lifetime risk for many other types of cancer such as in the stomach, small bowel, brain, skin, pancreas, biliary tract, ovary (only for women) and upper urinary tract [21]. Lynch syndrome carriers are predisposed to develop MSI cancers due to have an inherited pathogenic variant in one allele of the affected MMR genes MLH1 , MSH2 , MSH6 or PMS2 [18] passed down in the family from parent to child. Upon the second somatic hit inactivating the remaining allele, MMR deficiency manifests in the 136 affected cell [17]. DNA replication errors, especially those which occur at repetitive sequences (microsatellites consisting of a consecutive series of identical basepairs) cannot be corrected by the mismatch repair system. MMR deficiency leads to microsatellite instability. MMR deficiency can be an initiating or a secondary event in Lynch syndrome carcinogenesis. This is reflected by the hypothesis of three pathways responsible for colorectal carcinogenesis in Lynch syndrome [22] (see Fig 1): One pathway of carcinogenesis starts with adenoma formation, then MMR deficiency and cancer outgrowth; the second is initiated by MMR deficiency, then adenoma formation and cancer outgrowth; and the third shows MMR deficiency as initiating event and invasive cancer growth.” In this statement initially LS is considered one delineated entity, while thereafter it is declared to be four distinct syndromes caused by four different genes when deranged, and EPCAM is not mentioned as it is another gene causing epigenetic inactivation of MSH2 – the MSH2 gene is not ‘mutated’ when the pathogenic inherited variant is EPCAM. The four different genes have different capacity to correct replication errors in different structures in the genome. They behave differently in every demonstrated way to measure their activities and corresponding consequences when not functioning properly. It is not proper to lump them all together in one group in the current context.

In particular, it is to me unlikely that the triggering event in colon cancer associated with PMS2 is the PMS2 variant – in contrast to MLH1 and PMS2 carriers, PMS2 colon cancer may follow the CIN/APC pathway and effects of intervention indicate that these actually may be prevented by adenomectomy: in the upper leftmost part of Fig 1 PMS2-associated colon cancer may by and large follow the upper pathway. To which degree the same is true for MSH6 carriers are to me unclear. The paper is to me based on observations in carriers of pathogenic MLH1/MSH2 variants, and should indicate so both in title and text.

Reading the paper, I get the impression that a ‘linear’ tree-step model is commonly accepted as the ordinary carcinogenetic pathway. To me, this is not true. The most frequent cancer in women – breast cancer – and the most frequent inherited cancer in women inherited breast/ovarian cancer – is not: In contrast to colon cancer where the patient is considered cured if no spread at diagnosis and no recurrence after some years, recurrence rate for breast cancer never flattens out. In particular, BRCA1-associated breast/ovarian cancer is demonstrated to spread before a tumour is detectable – to the effect of clinical advice on prophylactic oophorectomy and mastectomy in healthy carriers. In the three-step model assumed by the authors, in these as frequent both sporadic and inherited cancers, the three step model is now recognized as false: Often, and especially so in BRCA1 carriers, the step causing spread (which should be the biological precursor of death) is very early. The breast cancer oncologists traditionally denote it ‘dormant breast cancer’. The discussion of over-diagnosis in breast cancer screening is commonly acknowledged, the corresponding in LS should be no surprise and a model to explain inherited cancer in path_MLH1/MSH2 carriers cannot ignore this. The problem with the LS discussion is that – in contrast to breast cancer screening – one assume cancer may be prevented by early diagnosis. This never was the goal of breast cancer screening, and has now been proven not to be true for LS as well. And, it is neither uniformly accepted to be true for sporadic colon cancer.

In short: The graphical image of path_MLH1/MSH2 carcinogenesis may be considered as opposing vectors in a multi-axis space, where the observed cancer incidence is the summary projection to one axis, and which is demonstrated to be a function of time between examinations to demonstrate cancer: over-diagnosis is a time-axis in the space. Without discussing this, the results presented give limited information.

The above argument are to indicate why ‘a linear model’ is not an obvious assumption to be true, over-diagnosis when colonoscopy is a demonstrated problem, the assumption of LS colorectal cancer to be preventable by frequent enough colonoscopy is not demonstrated to be true, while – in contrast to BRCA1 associated breast/ovarian cancer – early diagnosis has been proven to dramatically reduce mortality.

The arguments above are to illustrate that while I get the impression of the paper that cancer = death validating ‘cancer’ as a surrogate endpoint for death, this is to me not so. And, back to the start, the paper still has no definition of ‘cancer’: An adenoma with micro-invasive cell growth may be biologically very different from a metastatic tumour.

In short – the generalization of the concepts discussed above are to me not true, neither is the overall terminology LS when the paper describes some biological findings in in MLH1/MSH2 carriers.

Independently of the above, the assumptions: “we assume that all genes implements all mutational processes that are independent of each other, which is either due to a independence indicated by data or due to missing medical insight suggesting otherwise

Independence of the Processes We require that the processes are independent of each other. In the context of the example above, we interpret from the first assumption that all combinations of mutations in the different genes are possible (i.e., there are no mutations that prevent other mutations) and that there are no additional states. This also implies that the order in which mutations are accumulated is ignored.” are wrong. Mutations as events may be interrelated – in linkage terminology when estimating genetic distance between loci denoted ‘inference’. The example of today is a SarsCov2 virus variant having increased infection rate due to multiple ‘mutations’ and the ‘missing link’ as an evolutionary process is not demonstrated. The evolutionary demonstration is copies of large gene sequences (transposable elements), in the current context an obvious example is the copying of the basic immune system structure to multiple close-to identical genetic parts dispersed and specialized to have different functions as afferent and efferent pieces in the immune response, including LS cancers. These multi-topic multi-mutations events are well known in cancer, may in most cases cause cell death, but may extremely rarely cause viable cells with complex alterations in their genomes. In the very many cell divisions in MSI crypts in MLH1/MSH2 carriers, cancer may extremely rarely be caused by such events causing a viable cancer cell not to be identified and destroyed by the host immune system. Such a hypothesis is conform with current understanding of cancer and evolutionary genetics and in accordance with empirical observations in LS.

In sum, there are arguments to be discussed to interpret the conclusions. While the results may be correct, to me the mns is still not good enough discussing to which degrees the mechanisms described are the ‘causes’ of the cancers observed. The demonstration of the concepts in the paper to be true, is not validating that these are the major and most frequent causes neither of cancer nor of death. This is not about the algorithms and results per se, but how to discuss the assumptions and conclusions in a broader perspective. The mns would to me be more valuable and not invite the counter-arguments above if proactively discuss these in more details.

Reviewer #3: The manuscript entitled "Mathematical Modeling of Multiple Pathways in Colorectal Carcinogenesis using Dynamical Systems with Kronecker Structure" by Haupt and colleagues has been extensively revamped and reorganized, receiving most if not all suggestion this and the other reviewers have made. The revised manuscript is definitely clearer and easier to read (for instance, there is now an explicit distinction between assumptions made for the Lynch syndrome case, and those that are general, and so on).

That said, there is still the need to double check and proof read the text, since some sentences are still a little unclear, or require revision of the punctuation. An example for instance is at page 15/55 (in the manuscript version with highlighed changes), lines 341-342: "For other types of cancer [COMMA HERE] or once new medical insights are gathered, they can and should be adapted.

Finally, I think a main point still remains to be addressed (or at least explicitly discussed): mutations and genetic alterations in cancer occur "clustered" in pathways –– NB: here not intended as sequences of events leading to carcinogenesis, but as signaling and biological pathways regulating cell processes –– so it would be interesting to understand and discuss how the model can accommodate this concept. The Wnt pathway, for instance, can be affected via alternative hits (in APC, Axin, β-catenin, etc, ...), so one is left to wonder how this is (or would be) handled in the model (if handled at all).

In other words, is the assumption that the Wnt pathway (or the MMR-deficient process, or the RAS pathway) is altered as a whole –– hence the genes in said pathways are interchangeable –– or the genes in the model are individually considered?

For MMR-deficiency it is acknowledged that the mutations could be in different genes, what about the other pathways?

There are implications here, since assuming and parametrizing only using the APC gene for the whole Wnt pathway, for instance, does not exclude that when this gene is not mutated, other genes with the same effect (i.e., altering Wnt signaling) are indeed mutated. Hence the whole model might be somewhat incorrect. Given the methodological nature of the paper, this aspect could be simply discussed, not overselling the results obtained on the medical and biological side (since they could not be exactly correct...)

**Have all data underlying the figures and results presented in the manuscript been provided?**

Reviewer #1: Yes

Reviewer #3: Yes

PLOS authors have the option to publish the peer review history of their article (what does this mean?). If published, this will include your full peer review and any attached files.

Reviewer #1: **Yes: **Pål Møller

Reviewer #3: No

Figure Files:

Data Requirements:

Reproducibility:

References:

---

## [Decision Letter · Decision Letter 2]

16 Apr 2021

Dear Mrs Haupt,

We are pleased to inform you that your manuscript 'Mathematical Modeling of Multiple Pathways in Colorectal Carcinogenesis using Dynamical Systems with Kronecker Structure' has been provisionally accepted for publication in PLOS Computational Biology.

As you can see in the comments attached, Reviewer 1 has suggested minor editing to place your work in a future perspective, which you could consider. 

Best regards,

Jing Chen

Guest Editor

PLOS Computational Biology

Natalia Komarova

Deputy Editor

PLOS Computational Biology

Reviewer's Responses to Questions

**Comments to the Authors:**

Reviewer #1: I have carefully read the authors responses to my last comments. I think I rest my case: this is their way of seeing the topic addressed, which is of interest, and which I should not try to distort besides clearing up concepts and nomenclature the way I have tried to make their position not to be misunderstood.

I trust the authors to have mended the mns as they indicate in the response, and feel no need to use my time to re-read the mns in details once more and which would have delayed this response.

I may still have one last wish, which would be that the mns declares that they present one way of seeing while there may be others valid positions as well and which may not mutually dis-validate each others: Proving one theory to be right may not necessarily imply another theory to be wrong. I would be happy if the mns clearly state that, especially when dealing with LS which has shown us that most we short time ago thought was 'true', still is 'true' but there is another reality as well which is 'true' and possibly as important when considering LS. This would also serve proactively against critics now and in the future - one may assume more facts to modify the paradigms we believe in, there is no reason to assume that we today have reached the final step of knowledge. In short, to me the paper would be even more valuable and probably more robust if proactively mention that it presents but one way of putting in context what we currently know.

Again, thx for the opportunity to referee this mns - it has been an interesting ride.

Reviewer #3: This manuscript is covering a very specific problem in a very specific field and it would not be of interest to the general public.

**Have the authors made all data and (if applicable) computational code underlying the findings in their manuscript fully available?**

Reviewer #1: None

Reviewer #3: Yes

PLOS authors have the option to publish the peer review history of their article (what does this mean?). If published, this will include your full peer review and any attached files.

Reviewer #1: **Yes: **Pål Møller

Reviewer #3: No

---

## [Editor Report · Acceptance letter]

11 May 2021

PCOMPBIOL-D-20-01806R2 

Mathematical Modeling of Multiple Pathways in Colorectal Carcinogenesis using Dynamical Systems with Kronecker Structure

Dear Dr Haupt,

I am pleased to inform you that your manuscript has been formally accepted for publication in PLOS Computational Biology. Your manuscript is now with our production department and you will be notified of the publication date in due course.

With kind regards,

Katalin Szabo
